



# Recent changes in terrestrial water storage in the Upper Nile Basin: an evaluation of commonly used gridded GRACE products

**Mohammad Shamsudduha[1, 2], Richard G. Taylor[2], Darren Jones[3], Laurent Longuevergne[4], Michael Owor[5] and Callist Tindimugaya[6]**

[1]Institute for Risk and Disaster Reduction, University College London, UK

[2]Department of Geography, University College London, UK

[3]Centre for Geography, Environment and Society, University of Exeter, UK

[4]CNRS – UMR 6118 Géosciences Rennes, Université de Rennes 1, France

[5]Department of Geology & Petroleum Studies, Makerere University, Uganda

[6]Directorate of Water Resources Management, Ministry of Water & Environment, Uganda

Correspondence to: M. Shamsudduha (m.shamsudduha@ucl.ac.uk)

**Abstract**

GRACE (Gravity Recovery and Climate Experiment) satellite data monitor large-scale changes in total terrestrial water storage (ΔTWS) providing an invaluable tool where in situ observations are limited. Substantial uncertainty remains, however, in the amplitude of GRACE gravity signals and the disaggregation of ΔTWS into individual terrestrial water stores (e.g. groundwater storage). Here, we test the phase and amplitude of GRACE ΔTWS signals from 5 commonly-used gridded products (i.e., NASA's *GRCTellus*: CSR, JPL GFZ; JPL-Mascons; GRGS GRACE) using in situ data and modelled soil-moisture from the Global Land Data Assimilation System (GLDAS). The focus of this analysis is a large and accurately observed reduction in ΔTWS of 75 km$^3$ from 2004 to 2006 in Lake Victoria in the Upper Nile Basin. We reveal substantial variability in current GRACE products to quantify the reduction of ΔTWS in Lake Victoria that ranges from 68 km$^3$ (GRGS) to 50 km$^3$ and 26 km$^3$ for JPL-Mascons and





*GRCTellus*, respectively. Representation of the phase in ΔTWS in the Upper Nile Basin by
GRACE products varies but is generally robust with GRGS, JPL-Mascons and *GRCTellus*
(ensemble mean of CSR, JPL and GFZ time-series data) explaining 91 %, 85 %, and 77 % of the
variance, respectively, in in-situ ΔTWS.  Resolution of changes in groundwater storage (ΔGWS)
from GRACE ΔTWS is greatly constrained by both uncertainty in modelled changes in soil-
moisture storage (ΔSMS) and the low annual amplitudes in ΔGWS (e.g., 3.5 to 4.4 cm) observed
in deeply weathered crystalline rocks underlying the Upper Nile Basin. Our study highlights the
substantial uncertainty in the amplitude of ΔTWS that can result from different data-processing
strategies in commonly used, gridded GRACE products.
**Keywords:** GRACE products; terrestrial water storage; groundwater; hard-rock aquifers; Lake
Victoria; Lake Kyoga; Sub-Saharan Africa
**1.   Introduction**
Satellite measurements under the Gravity Recovery and Climate Experiment (GRACE) mission
have, since March 2002 (Tapley et al., 2004), enabled remote monitoring of large-scale (~200
000 km$^2$) spatio-temporal changes in total terrestrial water storage (ΔTWS) at 10-day to monthly
timescales (Longuevergne et al., 2013; Humphrey et al., 2016). Over the last 15 years, studies in
basins around the world (Rodell and Famiglietti, 2001; Strassberg et al., 2007; Leblanc et al.,
2009; Chen et al., 2010; Longuevergne et al., 2010; Frappart et al., 2011; Jacob et al., 2012;
Shamsudduha et al., 2012; Arendt et al., 2013; Kusche et al., 2016) show that GRACE satellites
trace natural (e.g., drought, floods, glaciers and ice melting, sea-level rise) and anthropogenic





(e.g., abstraction-driven groundwater depletion) influences on ΔTWS. GRACE-derived TWS
provides vertically-integrated water storage changes in all water-bearing layers (Wahr et al.,
2004; Strassberg et al., 2007; Ramillien et al., 2008) that include (Eq. 1) surface water storage in
rivers, lakes, and wetlands (ΔSWS), soil moisture storage (ΔSMS), ice and snow water storage
(ΔISS), and groundwater storage (ΔGWS). GRACE measurements have over the last decade
become an important hydrological tool for quantifying basin-scale ΔTWS (Güntner, 2008; Xie et
al., 2012; Hu and Jiao, 2015) and are increasingly being used to assess spatio-temporal changes
in specific water stores (Famiglietti et al., 2011; Shamsudduha et al., 2012; Jiang et al., 2014;
Castellazzi et al., 2016; Long et al., 2016; Nanteza et al., 2016) where time-series records of
other individual freshwater stores are available (Eq. 1).

$\Delta TWS_t = \Delta GWS_t + \Delta ISS_t + \Delta SWS_t + \Delta SMS_t$         (1)

GRACE-derived ΔTWS derive from monthly gravitational fields which can be represented as
spherical harmonic coefficients that are noisy as depicted in north-south elongated linear features
or "stripes" on monthly global gravity maps (Swenson and Wahr, 2006; Wang et al., 2016).
Post-processing of GRACE SH data is therefore required. The most popular GRACE products
are NASA's *GRCTellus* land gravity solutions (i.e., spherical harmonics based CSR, JPL and
GFZ), which require scaling factors to recover spatially smoothed TWS signals (Swenson and
Wahr, 2006; Landerer and Swenson, 2012). Additionally, NASA's new monthly gridded
GRACE product, Mass Concentration blocks (i.e., Mascons), estimate terrestrial mass changes
directly from inter-satellite acceleration measurements and can be used without further post-
processing (Rowlands et al., 2010; Watkins et al., 2015). GRGS GRACE are also spherical

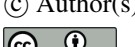



harmonic-based products available at a 10-day timestep and can also be used directly since
gravity fields are stabilised during the processing of GRACE satellite data (Lemoine et al., 2007;
Bruinsma et al., 2010).

Restoration of the amplitude of *GRCTellus* TWS data, dampened by spatial Gaussian filtering
with a large smoothing radius (e.g., 300 to 500 km), is commonly achieved using scaling factors
that derive from a priori model of freshwater stores, usually a global-scale Land-Surface Model
or LSM (Long et al., 2015). However, signal-restoration methods are emerging that do not
require hydrological model or LSM (Vishwakarma et al., 2016). Substantial uncertainty
nevertheless persists in the magnitude of applied scaling factors (e.g., *GRCTellus*) and
corrections (Long et al., 2015). In situ observations provide a valuable and necessary constraint
to the scaling of TWS signals over a particular study area as no consistent basis for ground-
truthing these factors exists.

The disaggregation of GRACE-derived ΔTWS anomalies into individual water stores (Eq. 1) is
commonly constrained by the limited availability of observations of terrestrial freshwater stores
(i.e., ΔSWS, ΔSMS, ΔGWS, ΔISS). Indeed, a major source of uncertainty in the attribution of
GRACE ΔTWS derives from the continued reliance on modelled ΔSMS derived from LSMs
(i.e., CLM, NOAH, VIC, MOSAIC) under the Global Land Data Assimilation System or
GLDAS (Rodell et al., 2004) and remote-sensing products (Shamsudduha et al., 2012; Khandu et
al., 2016). Further, analyses of GRACE-derived ΔGWS often assume ΔSWS is limited (Kim et
al., 2009) yet studies in the humid tropics and engineered systems challenge this assumption
showing that it can overestimate ΔGWS (Shamsudduha et al., 2012; Longuevergne et al., 2013).

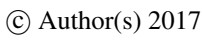



Robust estimates of ΔGWS from GRACE gravity signals have, to date, been developed in
locations where ΔSWS is well constrained by in situ observations and groundwater is used
intensively for irrigation so that ΔGWS comprises a significant (>10 %) proportion of ΔTWS
(Leblanc et al., 2009; Famiglietti et al., 2011; Shamsudduha et al., 2012; Scanlon et al., 2015). In
Sub-Saharan Africa (SSA), intensive groundwater withdrawals are restricted to a limited number
of locations (e.g., irrigation schemes, cities) and constrained by low-storage, low-transmissivity
aquifers in the  deeply weathered crystalline rocks that underlie ~40 % of this region
(MacDonald et al., 2012) including the Upper Nile Basin. Consequently, the ability of low-
resolution GRACE gravity signals to trace ΔGWS in these hard-rock environments is unclear. A
recent study (Nanteza et al., 2016) applies NASA's *GRCTellus* (CSR GRACE) data over large
basin areas (>300 000 km$^2$) of East Africa and argues that ΔGWS can be estimated with
sufficient reliability to characterise regional groundwater systems after accounting for ΔSWS by
satellite altimetry and ΔSMS data from the GLDAS LSM ensemble (Rodell et al., 2004).

Here, we exploit a large-scale reduction and recovery in surface water storage that was recorded
within Lake Victoria (Fig. 1), the world's second largest lake by surface area (67 220 km$^2$)
(UNEP, 2013) and eighth largest by volume (2 760 km$^3$) (Awange et al., 2008).  This well-
constrained reduction in ΔSWS comprises a decline in lake level of 1.2 m between May 2004
and February 2006, equivalent to a lake-water volume (ΔSWS) loss of 81 km$^3$ that resulted, in
part, from excessive dam releases (Fig. 2). We test the ability of current GRACE products to
represent the amplitude and phase of this voluminous and well-constrained change in freshwater
storage. Our analysis focuses on both the Lake Victoria Basin (hereafter LVB) (256 100 km$^2$)
and Lake Kyoga Basin (hereafter LKB) (79 270 km$^2$) (Fig. 1). Applying in situ observations of



ΔSWS and ΔGWS combined with simulated ΔSMS by the GLDAS LSMs, we assess: (1) the
ability of current gridded GRACE products (i.e., *GRCTellus*, JPL-Mascons, GRGS GRACE) to
measure a well constrained ΔTWS in the Upper Nile Basin from 2003 to 2012 focusing on the
unintended experiment within the LVB from 2004 to 2006; and (2) the sensitivity of a
disaggregated GRACE ΔTWS signals to trace ΔGWS in a deeply weathered crystalline rock
aquifer systems underlying the Upper Nile Basin.


**2.    The Upper Nile Basin**
**2.1    Hydroclimatology**
The Upper Nile Basin, the headwater area of the ~3 400 000 km$^2$ Nile Basin (Awange et al.,
2014), includes both the Lake Victoria Basin (LVB) and Lake Kyoga Basin (LKB). Mean annual
rainfall over the entire basin varies from 650 to 2900 mm (TRMM monthly rainfall; 2003−2012)
with an average of 1300 mm (σ=354 mm) (Fig. 3). Mean annual gauged rainfall at different
stations, Jinja, Bugondo and Entebbe measured is 1195, 1004 and 1541 mm, respectively (Owor
et al., 2011). Rainfall over Lake Victoria is typically 25−30 % greater than that measured in the
surrounding catchment (Fig. 3), which is partially explained by the nocturnal 'lake breeze' effect
(Yin and Nicholson, 1998; Nicholson et al., 2000; Owor et al., 2011).

Estimates of mean annual evaporation from the surface of Lake Victoria vary from 1260 mm
(UNEP, 2013) to 1566 mm (Hoogeveen et al., 2015) whereas mean annual evaporation from the
surface of Lake Kyoga is estimated to vary from 1205 mm (Brown and Sutcliffe, 2013) to 1660
mm (Hoogeveen et al., 2015). Evapotranspirative fluxes from the surrounding swamps in Lake





Kyoga are estimated to be much higher and approximately 2230 mm yr$^{-1}$ (Brown and Sutcliffe,

144 2013).


Annual rainfall is predominantly bimodal in distribution (Fig. 4) with two distinct rainy seasons
driven by the movement of the Intertropical Convergence Zone (ITCZ) (Awange et al., 2013).
Long rains (March to May) and short rains (September to November) account for approximately
40% and 25% of annual rainfall respectively (Basalirwa, 1995; Indeje et al., 2000). The latter
rainfalls are particularly influenced by El-Niño Southern Oscillation (ENSO) and Indian Ocean
Dipole (IOD). GRACE-derived ΔTWS within the LVB shows a statistical association ($R^2$) of
0.56 with ENSO and 0.48 with IOD (Awange et al., 2014).

**2.2 Lakes Victoria and Kyoga**
Located between 31°39' E and 34°53' E longitudes, and 0°20' N and 3°00' S latitudes, Lake
Victoria (Fig. 1) is located in Tanzania, Uganda and Kenya where each accounts for 51 %, 43 %
and 6 % of lake surface area respectively (Kizza et al., 2012). Lake Victoria is relatively shallow
with a mean depth of ~40 m and a maximum depth of 84 m (UNEP, 2013) akin to many shallow,
open surface-water bodies as well as permanent and seasonal wetlands occupying low relief
plateau across the Great Lakes Region of Africa (Owor et al., 2011). Moreover, the western and
northwestern lake bathymetry is characterised by even shallower depths of between 4 and 7 m
(Owor, 2010). Hydrologically, lake input is dominated by direct rainfall (84 % of total input); the
remainder derives primarily from river inflows as direct groundwater inflow (<1 %) is negligible
(Owor et al., 2011). Approximately 25 major rivers flow into Lake Victoria with a total
catchment area of ~194 000 km$^2$; the largest tributary, River Kagera, contributes ~30 % of total





river inflows (Sene and Plinston, 1994). Lake Victoria outflow to Lake Kyoga occurs at Jinja
(Fig. 1).

Lake Kyoga (Fig. 1), located between 32°10' E and 34°20' E longitudes, and 1°00' N and 2°00'
N latitudes, has a mean area of 1 720 km$^2$ with an estimated mean volume of 12 km$^3$ (Owor,
2010; UNEP, 2013). According to the recent global *HydroSHEDS* (Hydrological data and maps
based on shuttle elevation derivatives at multiple Scales) database, the Lake Kyoga has a total
surface area of 2 729 km$^2$ (Lehner et al., 2008). Lake Kyoga comprises lake-zone and flow-
through conduit areas. The lake zone in Lake Kyoga is very shallow with a mean depth of 3.5 to
4.5 m (Owor, 2010). Lake Kyoga has a through-flow channel (mean depth 7 to 9 m) where the
main Victoria Nile River flows (Owor, 2010) and acts as a linear reservoir with the annual water
balance predominantly governed by the discharge of the Victoria Nile from Lake Victoria. Lake
Kyoga has a through-flow channel (mean depth 7−9 m) where the main Victoria Nile River
flows (Owor, 2010). Whilst numerous rivers flow into Lake Kyoga (e.g. Rivers Mpologoma,
Awoja, Omunyal, Abalang, Olweny, Sezibwa and Enget) (Owor, 2010), the majority contributes
a fraction of their former volume upon reaching the lake (Krishnamurthy and Ibrahim, 2013)
due, in part, to evapotranspirative losses from fringe swamp areas (4 510 km$^2$) surrounding the
lake (UNEP, 2013).

**2.3   Hydrogeological setting**
The Upper Nile Basin is underlain primarily by deeply weathered crystalline rock aquifer
systems that have evolved through long-term, tectonically-driven cycles of deep weathering and
erosion (Taylor and Howard, 2000). Groundwater occurs within unconsolidated regoliths or





'saprolite' and, below this, in fractured bedrock, known as 'saprock'. Bulk transmissivities of the
saprolite and saprock aquifers are generally low (1 to 20 $m^2$ $d^{-1}$) (Taylor and Howard, 2000;
Owor, 2010) and field estimates of the specific yield of the saprolite, the primary source of
groundwater storage in these aquifer systems, are 2 % based on pumping-tests with tracers
(Taylor et al., 2010) and magnetic resonance sounding experiments (Vouillamoz et al., 2014).
Borehole yields are highly variable but generally low (0.5 to 20 $m^3$ $h^{-1}$) yet are of critical
importance to the provision of safe drinking water.

**2.4    An observed reduction in TWS in the LVB**
In 1954, the construction of the Nalubaale Dam (formerly Owen Falls Dam) at the outlet of Lake
Victoria at Jinja transformed the lake into a controlled reservoir (Sene and Plinston, 1994).
Operated as a run-of-river hydroelectric project to mimic pre-dam outflows, the 'Agreed Curve'
between Uganda and Egypt dictated dam releases that were controlled on a 10-day basis and
generally adhered to, with compensatory discharge releases to minimise any departures, until the
construction of the Kiira dam at Jinja in 2002 (Sene and Plinston, 1994; Owor et al., 2011).

The combined discharge of the Nalubaale and Kiira Dams enabled total dam releases (Fig. 2) to
substantially exceed the Agreed Curve (Sutcliffe and Petersen, 2007) and between May 2004 and
February 2006 the lake level dropped by 1.2 m (equivalent ΔSWS loss of 81 $km^3$) (Owor et al.,
2011). Mean annual releases were 1387 $m^3$ $s^{-1}$ (+162 % of Agreed Curve) in 2004 and 1114 $m^3$ $s^{-1}$
(+148 % of Agreed Curve) in 2005. Sharp reductions in dam releases in 2006 helped to arrest
and reverse the lake-level decline with lake levels stabilising by early 2007.





**3.    Data and Methods**
**3.1    Datasets**
We use publicly available time-series records of: (1) GRACE TWS solutions from a number of
data processing and dissemination centres including NASA's *GRCTellus* land solutions (RL05
for CSR, GFZ (version DSTvSCS1409), RL05.1 for JPL (version DSTvSCS1411), JPL-Mascons
solution (version RL05M_1.MSCNv01)), and the French National Centre for Space Studies
(CNES) GRGS (version GRGS RL03-v1); (2) NASA's Global Land Data Assimilation System
(GLDAS) simulated soil moisture data from 3 global land surface models (LSMs) (CLM,
NOAH, VIC); and (3) precipitation data from NASA's Tropical Rainfall Measuring Mission
(TRMM) satellite mission. We also employ in-situ observations of lake levels and groundwater
levels from a network of gauges and monitoring wells operated by the Ministry of Water and
Environment in Entebbe (Uganda). Datasets are described briefly below.

**3.1.1 Delineation of basin study areas**
Delineation of the Lake Victoria Basin (LVB) and Lake Kyoga Basin (LKB) was conducted in
Geographic Information System (GIS) under ArcGIS (v.10.3.1) environment using the
'Hydrological Basins in Africa' datasets derived from *HydroSHEDS* database (available at
http://www.hydrosheds.org/) (Lehner et al., 2006, 2008). Regional water bodies including Lakes
Victoria and Kyoga (Fig. 1) were spatially defined by the Inland Water dataset available globally
at country scale from DIVA-GIS (Hijmans et al., 2012). Computed areas of the basins and lake
surface areas are summarised in Table 1 along with previously estimated figures from other
studies.



### 3.1.2 GRACE-derived terrestrial water storage (TWS)


Twin GRACE satellites provide monthly gravity variations  interpretable as ΔTWS (Tapley et
al., 2004) with an accuracy of ~1.5 cm (Equivalent Water Thickness or Depth) when spatially
averaged (Wahr et al., 2006). In this study, we apply 5 different monthly GRACE solutions for
the period of January 2003 to December 2012: post-processed, gridded (1° × 1°) GRACE-TWS
time-series records from 3 *GRCTellus* land solutions from CRS, JPL and GFZ processing centres
(available at http://grace.jpl.nasa.gov/data) (Swenson and Wahr, 2006; Landerer and Swenson,
2012),  JPL-Mascons (Watkins et al., 2015; Wiese et al., 2015), and GRGS GRACE products
(CNES/GRGS release RL03-v1) (Biancale et al., 2006).

*GRCTellus* land datasets are post-processed from two versions, RL05 and RL05.1 of spherical
harmonics released by the University of Texas at Austin Centre for Space Research (CSR) and
the German Research Centre for Geosciences Potsdam (GFZ), and the NASA's Jet Propulsion
Laboratory (JPL) respectively. *GRCTellus* datasets are available at monthly timestep at a spatial
resolution of 1° × 1° grids (~111 km at equator).

Post-processing of *GRCTellus* GRACE datasets primarily involve (i) removal of atmospheric
pressure or mass changes based on the European Centre for Medium-Range Weather Forecasts
(ECMWF) model; (ii) a glacial isostatic adjustment (GIA) correction based on a viscoelastic 3-D
model of the Earth (Geruo et al., 2013); and (iii) an application a destriping filter plus a 300-km
Gaussian to minimise the effect of correlated errors (i.e., destriping) manifested by N-S
elongated stripes in GRACE monthly maps. However, the use of a large spatial filter and
truncation of spherical harmonics leads to energy removal so scaling coefficients or factors are





applied to the GRACE-derived TWS data in order to restore attenuated signals (Landerer and
Swenson, 2012). Dimensionless scaling factors are also provided as 1° x 1° bins that derive from
the Community Land Model (CLM4.0) (Landerer and Swenson, 2012).

*GRCTellus* JPL-Mascons (version RL05M_1.MSCNv01) data processing also involves a glacial
isostatic adjustment (GIA) correction based on a viscoelastic 3-D model of the Earth (Geruo et
al., 2013). JPL-Mascons applies no spatial filtering as JPL-RL05M directly relates inters-satellite
range-rate data to mass concentration blocks or Mascons to estimate global monthly gravity
fields in terms of equal area $3° \times 3°$ mass concentration functions to minimise measurement
errors. The use of Mascons and the special processing result in better signal-to-noise ratios of the
mascon fields compared to the conventional spherical harmonic solutions (Watkins et al., 2015).
For convenience, gridded Mascons fields are provided at a spatial sampling of 0.5° in both
latitude and longitude (~56 km at the equator). As with *GRCTellus* GRACE datasets the
neighbouring grid cells are not 'independent' of each other and cannot be interpreted
individually at the 1° or 0.5° grid scale (Watkins et al., 2015).

GRGS/CNES GRACE monthly products (version RL03-v1) are processed and made publicly
available (http://grgs.obs-mip.fr/grace) by the French Government space agency, National Centre
for Space Studies or Centre National d' Études Spatiales (CNES). The post-processing of GRGS
data involves taking into account of gravitational variations such as Earth tides, ocean tides, and
3D gravitational potential of the atmosphere and ocean masses (Bruinsma et al., 2010). The
remaining signals for time-varying gravity fields therefore represent changes in terrestrial
hydrology including snow cover, baroclinic oceanic signals and effects of post-glacial rebound

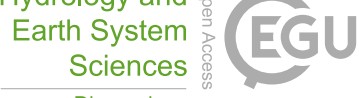



(Biancale et al., 2006; Lemoine et al., 2007). Further details on the Earth's mean gravity-field
models can be found on the official website of GRGS/LAGEOS (http://grgs.obs-mip.fr/grace/).

GRACE satellites were launched in 2002 to map the variations in Earth's gravity field over its 5-
year lifetime but both satellites are still in operation even after more than 14 years. However,
active battery management since 2011 has led the GRACE satellites to be switched off every 5−6
months for 4−5 week durations in order to extend its total lifespan (CSR, 2016). As a result,
GRACE ΔTWS time-series data have some missing records that are linearly interpolated
(Shamsudduha *et al*., 2012). In this study, we derive ΔTWS time-series data as equivalent water
depth (cm of $H_2O$) using the basin boundaries (GIS shapefiles) for masking the 1° × 1° grids.

**3.1.3 Soil moisture storage (SMS)**
NASA's Global Land Data Assimilation System (GLDAS) is an uncoupled land surface
modelling system that drives multiple land surface models (GLDAS LSMs: CLM, NOAH, VIC
and MOSAIC) globally at high spatial and temporal resolutions (3-hourly to monthly at 0.25° ×
0.25° grid resolution) and produces model results in near-real time (Rodell et al., 2004). These
LSMs provide a number of output variables which include soil moisture storage (SMS). Similar
to the approach applied in the analysis of GRACE-derived ΔTWS analysis in the Bengal Basin
(Shamsudduha et al., 2012), we apply simulated monthly ΔSMS records at a spatial resolution of
1° × 1° from 3 GLDAS LSMs: the Community Land Model (CLM, version 2) (Dai et al., 2003),
NOAH (version 2.7.1) (Ek et al., 2003) and the Variable Infiltration Capacity (VIC) model
(version 2.7.1) (Liang et al., 2003). The respective depths of modelled soil profiles are 3.4 m, 2.0
m, and 1.9 m in CLM (10 vertical layers), NOAH (4 vertical layers), and VIC (version 1.0) (3



vertical layers). Because of the absence of in situ soil moisture data in the study areas we apply
an ensemble mean of the aforementioned 3 LSMs-derived simulated ΔSMS time-series records
in order to disaggregate GRACE ΔTWS signals.

**3.1.4 Surface water storage (SWS)**
Daily time-series of ΔSWS are computed from in situ (gauged) lake-level observations at Jinja
for Lake Victoria and Bugondo for Lake Kyoga (Fig.s 1 and 2) compiled by the Ugandan
Ministry of Water and Environment (Directorate of Water Resources Management). Mean
monthly anomalies for the period of 2003–2012 were computed as an equivalent water depth
using Eq. (2). Missing data in the time series (2003–2012) records are linearly interpolated. For
instance, in case of monthly ΔSWS derived from Lake Kyoga water levels, there is one missing
record (December 2005).

$\Delta SWS = \Delta Lake\ Level \times \left( \frac{Lake\ Area}{Total\ Basin\ Area} \right)$ (2)

**3.1.5 Groundwater storage (GWS)**
Time series of ΔGWS are constructed from in situ piezometric records from 6 monitoring wells
located in LVB and LKB where near-continuous, daily observations exist from 2003 to 2012 and
have been compiled by the Ugandan Ministry of Water and Environment (Directorate of Water
Resources Management) (Owor et al., 2009; Owor et al., 2011). Monitoring boreholes were
installed into weathered, crystalline rock aquifers that underlie much of LVB and LKB, and are
remote from local abstraction. As such, they represent variations in groundwater storage



influenced primarily by climate variability. Mean monthly anomalies of ΔGWS, normalised to
2003−2012, were derived from near-continuous, daily observations at Entebbe, Rakai and
Nkokonjeru for LVB and at Apac, Pallisa and Soroti for LKB (Fig. 1; Table 2). These time series
data are a sub-set of the total number of available monitoring-well records in the LVB and LKB
following a rigorous review of groundwater-level records conducted at a dedicated workshop at
the Ministry of Water & Environment in January 2013. These records represent shallow
groundwater-level observations within the saprolite that is dynamically connected to surface
waters (Owor et al. 2011). The limited spatial coverage in quality-controlled piezometry,
especially for the LVB, represents an important limitation in our analysis. Mean monthly
anomalies were translated into an equivalent water depth (Eq. 3) by applying a range of specific
yield ($S_y$) values (1−6 % with an average of 3 %) although estimates of $S_y$ in hard-rock
environments are observed to vary from < 2% to 8 % (Taylor et al., 2010; Taylor et al., 2013;
Vouillamoz et al., 2014) using Eq. (3). Missing data in the time series were linearly interpolated.
In case of monthly ΔGWS that derived from borehole (n=6) observations, missing records range
from 1−9 months (120 months in 2003−2012) with three boreholes (Soroti, Rakai and
Nkonkonjero) with time-series records ending in June−July 2010.

$$\Delta GWS = \Delta h * S_y * \left( \frac{Land\ Area}{Total\ Basin\ Area} \right) \qquad (3)$$

### 3.1.6 Rainfall data

We apply Tropical Rainfall Measuring Mission (TRMM) (Huffman et al., 2007) monthly
product (3B43 version 7) for the period of 2003 to 2012 at 0.25° × 0.25° spatial resolution and
aggregate to 1° × 1° grids over LVB and LKB. General climatology of the Upper Nile Basin is




represented by long-term (2003–2012) mean annual rainfall (Fig. 3) and seasonal rainfall pattern
(Fig. 4). TRMM rainfall measurements show a good agreement with limited observational
precipitation records (Awange et al., 2008; Awange et al., 2014).


**3.2  Methodologies**
**3.2.1  GRACE ΔTWS estimation**
First, the 1° × 1° gridded monthly anomalies of GRACE-derived ΔTWS and GLDAS LSMs
derived ΔSMS are masked over the area of LVB and LKB (see supplementary Fig. S1). GRACE
ΔTWS along with GLDAS ΔSMS are extracted for the marked 1° × 1° grid cells for LVB and
LKB and the grid values are spatially aggregated to form time-series of monthly anomalies
ΔTWS and ΔSMS. Second, scaling coefficients or factors provided at 1° × 1° grids are applied to
each corresponding GRACE ΔTWS grids for NASA's *GRCTellus* products only in order to
restore attenuated signals during the post-processing using Eq. (4) (Landerer and Swenson,
2012). We apply an ensemble mean GRACE ΔTWS of 3 *GRCTellus* gridded products (i.e., CSR,
GFZ, and JPL solutions) as our exploratory analyses reveal that the time-series records over the
Lake Victoria Basin are highly correlated ($r >0.95$, $p$-value $<0.001$) and the Root Mean Square
Error (RMSE) is very small (ranges from 1.3 to 1.9 cm) among the time-series records.

$g^1(x, y, t) = g(x, y, t) \times s(x, y)$ $\hspace{3cm}$ (4)

Here, $g^1(x, y, t)$ represents each un-scaled grid where $x$ represents longitude, $y$ represents
latitude, and $t$ represents time (month), and $s(x, y)$ is the corresponding scaling factor.




### 3.2.2 GRACE ΔTWS reconciliation

Reconciling GRACE-derived TWS with ground-based observations is limited by the paucity of
in situ observations of SMS, SWS and GWS in many environments. In addition, direct
comparisons between in situ observations of ΔSMS, ΔSWS and ΔGWS and gridded GRACE
ΔTWS anomalies are complicated by substantial differences in spatial scales, which need to be
considered prior to analysis (Becker et al., 2010). The disaggregation of GRACE ΔTWS into
individual water store can also propagate errors to disaggregated components. Here, we construct
in situ ΔTWS (i.e., combined signals of ΔSMS, ΔSWS and ΔGWS) for the Lake Victoria Basin
and attempt to reconcile with GRACE-derived ΔTWS. One feature of GRACE ΔTWS among the
3 solutions we apply in this study is the considerable variation in amplitudes that exist over the
period of 2003 to 2012.  In addition, for the *GRCTellus* products, we conduct scaling
experiments, outlined below, to both the ensemble GRACE ΔTWS and in situ ΔSWS in an
attempt to reconcile satellite and in situ measures.

Firstly, *GRCTellus* GRACE ΔTWS gridded data are generally scaled up using dimensionless
gridded scaling factors that are provided separately and are independent of ΔTWS grids
(Landerer and Swenson, 2012). A number of GRACE studies (Rodell et al., 2009; Sun et al.,
2010; Shamsudduha et al., 2012) around the world have applied scaling factors in three different
ways: (1) single scaling factor based on regionally averaged time series, (2) spatially distributed
or gridded scaling factors based on time-series at each grid point, and (3) gridded-gain factors
estimated as a function time or of temporal frequency (Landerer and Swenson, 2012; Long et al.,
2015). In this study, we apply the gridded scaling factors approach to adjust ΔTWS time-series





records. For a further experiment, we apply a basin-averaged scaling factor ranging from 1.1 to
2.0 and employ RMSE to assess their relative performance. With reference to GRACE ΔTWS
and in situ ΔTWS relationship, the scaling factor producing the lowest RMSE is applied.

Secondly, in the LVB, ΔSWS is the largest contributor to ΔTWS. GRACE ΔTWS analyses
commonly apply the same scaling factor as ΔTWS to all other individual components (Landerer
and Swenson, 2012). We apply spatially-averaged scaling factors representative of (1) Lake
Victoria and its surrounding grid cells (experiment 1: s=0.71; range 0.02−1.5), and (2) the open-
water surface of Lake Victoria without surrounding grid cells (experiment 2: s=0.11; range
0.02−0.30). In addition, we also apply a spatially-averaged scaling factor (s=0.39; range
0.03−1.48) to JLP-Mascons signal to adjust the in-situ ΔSWS.


**4.    Results**
Monthly time-series records (January 2003 to December 2012) are presented in Figures 5 and 6
respectively for Lake Victoria Basin (LVB) and Lake Kyoga Basin (LKB) of (a) GRACE ΔTWS
from *GRCTellus* GRACE ΔTWS (ensemble mean of CSR, GFZ, and JPL solutions), GRGS and
JPL-Mascons, (b) GLDAS land surface models (LSMs) derived ΔSMS (ensemble mean of 3
LSMs: NOAH, CLM, VIC), (c) in situ ΔSWS from lake levels records, and (d) in situ ΔGWS
borehole observations. Monthly rainfall derived from TRMM satellite observations over the
same period are shown on the bottom panel (d). Time-series records of all ΔTWS components
and rainfall are aggregated for LVB to represent the average seasonal (monthly) pattern of each





signal (Fig. 4) that shows an obvious lag (~1 month) between peak rainfall (March−April) and
ΔTWS and its individual components.

Mean annual (2003−2012) amplitudes of various GRACE-derived ΔTWS signals, in situ ΔTWS,
ensemble mean of simulated ΔSMS, in situ ΔSWS and ΔGWS time-series records (Figs. 5 and 6)
are calculated (see supplementary Table S1) for both LVB and LKB. Mean annual amplitude of
GRACE ΔTWS ranges from 11.7 to 20.6 cm among *GRCTellus*, GRGS and JPL-MASCON
GRACE products in LVB, and from 8.4 to 16.4 respectively in LKB. Mean annual amplitude of
in situ ΔSWS is much greater (14.8 cm) in LVB than in LKB (3.8 cm). GLDAS LSMs derived
ensemble mean ΔSMS amplitude in LVB is 7.9 cm and 7.3 cm in LKB. The standard deviation
in ΔSMS varies substantially in LVB (1.2 cm, 4.2 cm, and 2.9 cm) LKB (1.3 cm, 4.7 cm, and 4.0
cm) for CLM, NOAH, and VIC models respectively. Mean annual amplitude of in situ ΔGWS
ranges from 4.4 cm (LVB) to 3.5 cm (LKB).

Time-series correlation (Pearson) analysis over various periods of interests (decadal: 2003–2012;
well-constrained SWS reduction or a period of unintended experiment: 2003–2006; controlled
dam operation: 2007–2012) reveals that GRACE-derived ΔTWS signals are strongly correlated
in both LVB and LKB (see supplementary Figs. S2–S7). For example, in LVB, in situ ΔSWS
shows a statistically significant (*p*-value <0.001) strong correlation (*r*=0.77–0.92) with all
GRACE- ΔTWS time-series (2003–2012) records. Similarly, simulated ΔSWS shows
statistically significant (*p*-value <0.001) strong correlation (*r*=0.72–0.78) with ΔTWS time-series
records. In contrast, in situ ΔGWS shows statistically significant (*p*-value <0.001) but moderate
correlation (*r*=0.46–0.56) with ΔTWS time-series records. Correlation among the variables



shows similar statistical associations for the periods of unintended experiment (2003–2006) and
controlled dam operation (2007–2012). In LKB, however, correlation among in situ ΔSWS and
GRACE ΔTWS time-series records is statistically significant ($p$-value <0.001) but poor in
strength ($r$=0.28–0.34). In situ ΔGWS shows statistically significant ($p$-value <0.001) moderate
correlation ($r$=0.40–0.47) with GRACE ΔTWS time-series records.

Time-series records of all 5 GRACE ΔTWS and in situ ΔTWS time-series records in both LVB
and LKB are shown in Figure 7 and results of temporal trends are summarised in Table 3.
Statistically significant ($p$-value <0.05) declining trends (–4.1 to –11.0 cm yr$^{-1}$ in LVB; –2.1 to –
5.6 cm yr$^{-1}$ in LKB) are consistently observed during the period of 2004 to 2006. Trends are all
positive in GRACE ΔTWS and in situ ΔTWS time-series records over the recent period of
controlled dam operation (2007–2012) in both LVB and LKB. Therefore, the overall, decadal
(2003–2012) trends are slightly rising (0.04 to 0.79 cm yr$^{-1}$) in LVB but nearly stable (–0.01 cm
yr$^{-1}$) in *GRCTellus* ΔTWS and slightly declining (–0.56 cm yr$^{-1}$) in situ ΔTWS over LKB. In
addition, short-term volumetric trends (2004–2006) in GRACE and in situ ΔTWS as well as
simulated ΔSMS and in situ ΔSWS are declining whereas in situ ΔGWS and rainfall anomalies
show slightly rising trends over the same period in LVB (see supplementary Figs. S8–S9).
Similar trends are reported in various signals over LKB but magnitudes are much smaller
compared to that of LVB, which is 3 times larger than LKB. Volumetric declines in ΔTWS in the
LVB for the period 2004 to 2006 are: 75 km$^3$ (in situ), 68 km$^3$ (GRGS), 50 km$^3$ (JPL-Mascons),
and 26 km$^3$ (*GRCTellus* ensemble mean of CRS, JPL and GFZ products).





Linear regression reveals that the association between GRACE-derived ΔTWS and in situ ΔTWS
is stronger in LVB ($R^2$=0.77−0.91) than in LKB ($R^2$=0.49−0.55) (see supplementary Table S1).
GRACE ΔTWS is unable to explain natural variability in in situ ΔTWS in LKB though this may
be explained by the fact that SWS in Lake Kyoga is influenced by dam releases from LVB.
Multiple linear regression analysis reveals that the relative proportion of variability in in situ
ΔTWS time-series record can be explained by ΔSWS (88.9 %), ΔSMS (9.4 %) and ΔGWS (1.9
%) in LVB; and by 37.2 %, 55.9 % and 6.9 % respectively in LKB. These results are indicative
only as these percentages can be biased by the presence of strong correlation among variables
and the order of these variables listed as predictors in the regression model.

Disaggregation of ΔGWS from GRACE ΔTWS time-series record from each product has been
carefully considered. In case of LVB, we apply a spatially-averaged multiplicative scaling factor
(1.7) to *GRCTellus* GRACE-derived ΔTWS dataset to amplify the signal that is better reconciled
with in situ ΔTWS (see supplementary Fig. S10). Additionally, for both *GRCTellus* and JPL-
Mascons ΔTWS disaggregation to ΔGWS a scaled down signal of in situ ΔSWS is applied.
Time-series record (2003−2012) of in situ ΔGWS in LVB weakly correlates ($r$=0.29, $p$-value
<0.001) with both *GRCTellus* and JPL-Mascons GRACE derived ΔGWS but shows no
correlation with GRGS ΔGWS (Fig. 8).

In LKB, in situ ΔGWS time-series record shows weaker and statistically insignificant correlation
($r$=0.16−0.19, $p$-value <0.08) with JPL-Mascons and GRGS GRACE-derived ΔGWS but shows
no correlation with *GRCTellus* ΔGWS (see supplementary Fig. S11). Furthermore, RMSE
among various GRACE-derived estimates of ΔGWS and in situ ΔGWS ranges from 3.0 cm





(GRACE ensemble), 3.7 cm (GRGS) to 6.4 cm (JPL-Mascons) in LVB, and from 3.4 (GRACE
ensemble), 5.6 cm (GRGS) to 6.8 cm (JPL-Mascons) in LKB.

**5.    Discussion**
We apply 5 different gridded GRACE products (*GRCTellus* – CSR, JPL and GFZ; GRGS and
JPL-Mascons) to test ΔTWS signals for in the Lake Victoria Basin (LVB) comprising a large and
accurately observed reduction (75 km$^3$) in ΔTWS from 2004 to 2006. Our analysis reveals that
all GRACE products capture this substantial reduction in terrestrial water mass but the
magnitude of GRACE ΔTWS among GRACE products varies substantially. For example,
*GRCTellus* underrepresents greatly (66 %) the reduction in in situ ΔTWS whereas GRGS
GRACE product underrepresents slightly (10 %). Over a longer period (2003−2012) in the
Upper Nile Basin, all GRACE products correlate well with in situ ΔTWS but, similar to the
unintended experiment, variability in amplitude is considerable (Fig. 9). The average amplitude
of ΔTWS is substantially dampened (i.e., 86 % less than in situ ΔTWS) in *GRCTellus* GRACE
products relative to GRGS (6 %) and JPL-Mascons (7 %) products in the LVB.

The 'true' amplitude in *GRCTellus* ΔTWS signal is generally reduced during the post-processing
of GRACE spherical harmonic fields, primarily due to spatial smoothing by a large-scale (e.g.,
300 km) Gaussian filter and truncation of gravity fields at a higher (degree 60 = 300 km) spectral
degree (Swenson and Wahr, 2006; Landerer and Swenson, 2012). Despite the application of
scaling coefficients based on CLM v.4.0 to amplify *GRCTellus* ΔTWS amplitudes at individual
grids, the basin-averaged (LVB) time-series record represents only 77 % variability in in situ



ΔTWS. Scaling experiments conducted here reveal that *GRCTellus* ΔTWS requires an additional
multiplicative factor of 1.7 in order to match in situ ΔTWS with a minimum RMSE (5.8 cm). On
the other hand, NASA's new gridded GRACE product, JPL-Mascons, that applies a priori
constraint in space and time to derive monthly gravity fields and undergoes some degree of
spatial smoothing (Watkins et al., 2015), represents nearly 85 % variability in in-situ ΔTWS. In
contrast, GRGS GRACE product, although applying truncation at degree 80 (~250 km), does not
suffer from any large-scale spatial smoothing and, is able to represent well (92 %) the variability
in in situ ΔTWS in the LVB.

A priori corrections of *GRCTellus* ensemble mean GRACE signals using a set of LSM-derived
scaling factors (i.e., amplitude gain) can lead to substantial uncertainty in ΔTWS (Long et al.,
2015). We show that the amplitude of simulated terrestrial water mass over the Upper Nile
Basins varies substantially among various LSMs (see supplementary Fig. S12). Most of these
LSMs (GLDAS models: CLM, NOAH, VIC) do not include surface water or groundwater
storage (Scanlon et al., 2012). Although CLM (v.4.0 and 4.5) includes a simple representation
(i.e., shallow unconfined aquifer) of groundwater (Niu et al., 2007; Oleson et al., 2008), it does
not consider recharge from irrigation return flows. In addition, many of these LSMs do not
consider lakes and reservoirs and, most critically, LSMs are not reconciled with in situ
observations. As a result, methods of rescaling the amplitude of GRACE signals based on a
priori information from LSMs contribute uncertainty to TWS signals.

The combined measurement and leakage errors, $\sqrt{bias^2 + leak^2}$ (Swenson and Wahr, 2006)
for *GRCTellus* ΔTWS based on CLM4.0 model for LVB and LKB are 7.2 cm and 6.6 cm



respectively. These values, however, do not represent mass leakage from the lake to the
surrounding area within the basin itself. A sensitivity analysis of *GRCTellus* and GRGS signals
for leakage from the lake into the basin area shows that leakage from Lake Victoria to LVB for
*GRCTellus* is substantially greater than GRGS product by a factor of ~2.6. In other words, 1 mm
change in the level of Lake Victoria represents an equivalent change of 0.12 mm in ΔTWS in
LVB for *GRCTellus* compared to 0.32 mm for GRGS. Consequently, changes in the amplitude
of GRGS ΔTWS are much greater (~38 %) than *GRCTellus*. During the observed reduction in
ΔTWS (75 km$^3$) from 2004 to 2006, the computed amplitude for GRGS is 68 km$^3$ whereas it is
26 km$^3$ for *GRCTellus*.

Another source of uncertainty that contributes toward ΔTWS anomalies in GRACE analysis is
the choice of simulated ΔSMS from various global-scale LSMs (e.g., Shamsudduha et al., 2012;
Scanlon et al., 2015). For example, the mean annual (2003−2012) amplitudes in simulated ΔSMS
in GLDAS LSMs (CLM, NOAH, VIC) vary substantially in LVB (3.5 cm, 10.2 cm, and 10.5
cm) and LKB (3.7 cm, 10.6 cm, and 7.7 cm) respectively. Due to an absence of a dedicated
monitoring network for soil moisture in the Upper Nile Basin, this study like many other
GRACE studies, is resigned to applying simulated ΔSMS from multiple LSMs arguing that the
use of an ensemble mean minimises the error associated with ΔSMS (Rodell et al., 2009).

Computed contributions of ΔGWS to ΔTWS in the Upper Nile Basins are low (<10 %).
GRACE-derived estimates of ΔGWS from all three products (*GRCTellus*, GRGS and JPL-
Mascons) correlate very weakly with in situ ΔGWS in both LVB and LKB. One curious
observation in LVB during the unintended experiment (2005−2006) is that in situ ΔGWS rises





whereas in situ ΔSWS and simulated ΔSMS decline. The available evidence in groundwater-
level records (e.g., Entebbe, Uganda) suggests that rainfall-generated groundwater recharge led
to an increased in ΔGWS while dam releases exceeding the "Agreed Curve" continued to reduce
ΔSWS (Owor et al., 2011).

Uncertainties in the estimation of GRACE-derived ΔGWS remain in: (i) the choice of scaling
factors applied to in situ ΔSWS associated with the disaggregation of ΔTWS from JPL-Mascons
and *GRCTellus* GRACE products, (ii) simulated ΔSMS by GLDAS land surface models, (iii) the
very limited spatial coverage in piezometry to represent in situ ΔGWS, and (iv) applied $S_y$ (3 %
with range from 1 % to 6 %) to convert in situ groundwater levels to ΔGWS. The lack of any
correlation in GRGS and in situ ΔGWS time-series records indicates that the magnitude of
uncertainty is larger than the overall variability in ΔGWS in low-storage, low-transmissivity
weathered crystalline aquifers within the Upper Nile Basin. In contrast to the assertions of
Nanteza et al. (2016) applying the *GRCTellus* CSR solution, we find that this uncertainty
prevents robust resolution of ΔGWS from GRACE ΔTWS in these complex hydrogeological
environments of East Africa. Despite substantial efforts to improve groundwater-level
monitoring[1] and to collate existing groundwater-level records[2] across Africa, we recognise that
understanding of in situ ΔGWS remains greatly constrained by limitations in current
observational networks and records. Since present uncertainties and limitations identified in the
Upper Nile Basin occur in many of the weathered hard-rock aquifer environments that underlie
40% of Sub-Saharan Africa (MacDonald et al., 2012), tracing of ΔGWS using GRACE in these
areas is unlikely to be robust until these uncertainties and limitations are better constrained.

---

[1] UPGro programme: https://upgro.org/
[2] The Chronicles Consortium: https://www.un-igrac.org/special-project/chronicles-consortium


## 6. Conclusions

The analysis of a large, accurately recorded reduction in the volume of Lake Victoria ($\Delta$SWS=81

km$^3$) from 2004 to 2006 exposes substantial variability among commonly-used 5 gridded

GRACE products (*GRCTellus* CSR, JPL, GFZ; GRGS; JPL-Mascons) to quantify the amplitude

of changes in terrestrial water storage ($\Delta$TWS). For this event, we estimate an overall decline in

'in situ' $\Delta$TWS (i.e., in situ $\Delta$SWS and $\Delta$GWS; simulated $\Delta$SMS) over the Lake Victoria Basin

(LVB) of 75 km$^3$. This value compares favourably with GRGS GRACE $\Delta$TWS (68 km$^3$), is

underrepresented by JPL-Mascons GRACE $\Delta$TWS (50 km$^3$), and is substantially

underrepresented by the ensemble mean of *GRCTellus* GRACE $\Delta$TWS (26 km$^3$). Attempts to

better reconcile *GRCTellus* GRACE $\Delta$TWS to in situ $\Delta$TWS through scaling techniques are

unable to represent adequately the observed amplitude in $\Delta$TWS.

From 2003 to 2012, GRGS, JPL-Mascons and *GRCTellus* GRACE products trace well the phase

in in situ $\Delta$TWS in the Upper Nile Basin that comprises both the LVB and Lake Kyoga Basin

(LKB). In the LVB for example, each explains 91 % (GRGS), 85 % (JPL-Mascons), and 77 %

(*GRCTellus* ensemble mean of CSR, JPL and GFZ) of the variance, respectively, in in situ

$\Delta$TWS. The relative proportion of variability in in situ $\Delta$TWS (variance 120 cm$^2$ LVB, 24 cm$^2$

LKB) is explained by in situ $\Delta$SWS (89 % LVB; 37 % LKB), GLDAS ensemble mean $\Delta$SMS (9

% LVB; 56 % LKB) and in situ $\Delta$GWS (2 % LVB; 7 % LKB); these percentages are indicative

as individual TWS components are strongly correlated. In situ $\Delta$GWS contributes minimally to

$\Delta$TWS and is only moderately associated with $\Delta$TWS ($r$=0.57, *p*-value <0.001). Resolution of

$\Delta$GWS from GRACE $\Delta$TWS in the Upper Nile Basin relies upon robust measures of $\Delta$SWS and

$\Delta$SMS; the former is observed in situ whereas the latter is limited by uncertainty in simulated

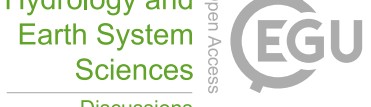

ΔSMS, represented here and in many GRACE studies by an ensemble mean of GLDAS LSMs.
Mean annual amplitudes in observed ΔGWS (2003–2012) from limited piezometry for the low-
storage and low-transmissivity aquifers in deeply weathered crystalline rocks that underlie the
Upper Nile Basin are small (3.5 to 4.4 cm for $S_y$= 0.03) and, given the current uncertainty in
simulated ΔSMS, are beyond the limit of what can be reliably quantified using current GRACE
satellite products.

Our examination of a large, mass-storage change (2004 to 2006) observed in the Lake Victoria
Basin highlights substantial variability in the measurement of ΔTWS using different gridded
GRACE products. Although the phase in ΔTWS is generally well recorded by all tested GRACE
products, substantial differences exist in the amplitude of ΔTWS that also influence the
disaggregation of individual terrestrial stores (e.g., groundwater storage) and estimation of trends
in TWS and individual, disaggregated freshwater stores. We note that the stronger filtering of the
large-scale (~300 km) gravity signal associated with *GRCTellus* results in greater signal leakage
relative to GRGS and JPL-Mascons. As a result, greater rescaling is required to resurrect signal
amplitudes in *GRCTellus* relative to GRGS and JPL-Mascons and these scaling factors depend
upon uncertain and incomplete a priori knowledge of terrestrial water stores derived from large-
scale models, which generally do not consider the existence of Lake Victoria, the second largest
lake by area in the world.



**Author contribution**


RT conceived this study for which preliminary analyses were carried out by DJ and MS. MS and
DJ have processed GRACE and all observational datasets and conducted statistical analyses and
GIS mapping. LL conducted the analysis of spatial leakage and bias in GRACE signals. CT, RT
and MO helped to establish, collate and analyse groundwater-level data; CT provided dam
release data. MS and RT wrote the manuscript and LL, DJ, MO and CT commented on draft
manuscripts.

**Competing interests**


The authors declare that they have no conflict of interest.

**Acknowledgements**


We kindly acknowledge NASA's MEaSUREs Program (http://grace.jpl.nasa.gov) for the freely
available gridded *GRCTellus* and JPL-MASCON GRACE data and French National Centre for
Space Studies (CNES) for GRGS GRACE data. NASA's Precipitation Processing Centre and
NASA's Hydrological Sciences Laboratory and the Goddard Earth Sciences Data and
Information Services Centre (GES DISC) are duly acknowledged for TRMM rainfall and soil
moisture data from GLDAS Land Surface Models. We kindly acknowledge the Directorate of
Water Resources Management in the Ministry of Water and Environment (Uganda) for the
provision of piezometric and lake-level data. Support from the UK government's UPGro
Programme, funded by the Natural Environment Research Council (NERC), Economic and
Social Research Council (ESRC) and the Department For International Development (DFID)
through the *GroFutures: Groundwater Futures in Sub-Saharan Africa* catalyst NE/L002043/1)
and consortium (NE/M008932/1) grant awards, is gratefully acknowledged.



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



**Figure Captions**

**Figure 1.** Map of the study area encompassing the Lake Victoria Basin (LVB) and Lake Kyoga
Basin (LKB), and location of the in situ monitoring stations. The Upper Nile Basin is marked by
a rectangle (red) within the entire Nile River Basin shown as a shaded relief index map.
Piezometric monitoring (red circles) and lake-level gauging (dark blue squares) stations are
shown on the map.

**Figure 2.** Observed daily total dam releases (blue line) and the agreed curve (red line) at the
outlet of Lake Victoria in Jinja from November 2007 to July 2009 (Owor et al., 2011).

**Figure 3.** Mean annual rainfall for the period of 2003−2012 derived from TRMM satellite
observations. Greater annual rainfall is observed over much of the Lake Victoria and
northeastern corner of the Lake Victoria Basin.

**Figure 4.** Seasonal pattern of TRMM-derived monthly rainfall, various GRACE-derived ΔTWS
signals [GRCE=ensemble mean of CSR, GFZ, and JPL; GRGS and JPL-Mascons (MSCN)
products], GLDAS LSMs ensemble ΔSMS, in situ ΔSWS and ΔGWS over the Lake Victoria
Basin.

**Figure 5.** Monthly time-series datasets for the Lake Victoria Basin (LVB) from January 2003 to
December 2012: (a) *GRCTellus* GRACE-derived ΔTWS (ensemble mean of CSR, GFZ, and
JPL), GRGS and JPL-Mascons ΔTWS time-series data; (b) GLDAS-derived ΔSMS (ensemble
mean of NOAH, CLM, and VIC); (c) lake-level-derived ΔSWS; and (d) borehole-derived ΔGWS
time-series data.

**Figure 6.** Monthly time-series datasets for the Lake Kyoga Basin (LKB) from January 2003 to
December 2012: (a) *GRCTellus* GRACE-derived ΔTWS (ensemble mean of CSR, GFZ, and
JPL), GRGS and JLP-Mascons ΔTWS time-series data; (b) GLDAS-derived ΔSMS (ensemble
mean of NOAH, CLM, and VIC); (c) lake-level-derived ΔSWS; and (d) borehole-derived ΔGWS
time-series data.





**Figure 7.** Comparison among time-series records of ΔTWS from *GRCTellus* (ensemble mean of
CSR, GFZ, and JPL), GRGS and JPL-MASCON GRACE products and in situ ΔTWS for the
Lake Victoria Basin (LVB) (a) and Lake Kyoga Basin (LKB), (b) for the period of 2003 to 2012.
The vertical grey lines represent monthly rainfall anomalies in LVB and LKB.

**Figure 8.** Estimates of in situ ΔGWS and GRACE-derived ΔGWS time-series records
(2003−2012) in LVB show a substantial variations among themselves. Note that an adjusted
ΔSWS (scaling factor of 0.11) is applied in the disaggregation of ΔGWS using *GRCTellus*
GRACE (ensemble mean of CSR, GFZ, and JPL) product; similarly, an adjusted ΔSWS (scaling
factor of 0.39) is applied for the JPL-Mascons product.

**Figure 9.** Taylor diagram shows strength of statistical association, variability in amplitudes of
time-series records and agreement among the reference data, in situ ΔTWS and *GRCTellus*
GRACE-derived ΔTWS (ensemble mean of CSR, GFZ, and JPL, GRGS and Mascons ΔTWS
time-series records), simulated ΔSMS (ensemble mean of NOAH, CLM, and VIC), in situ
ΔSWS, and in situ ΔGWS over the LVB. The solid arcs around the reference point (black
square) indicate cantered Root Mean Square (RMS) differences among in situ ΔTWS and other
variables, and the dashed arcs from the origin of the diagram indicate variability in time-series
records. Data for Lake Victoria Basin (LVB) are only shown in this diagram.





**Table 1.** Estimated areal extent (km$^2$) of the Lake Victoria Basin (LVB), Lake Kyoga Basin
(LKB), Lake Victoria and Lake Kyoga.

| Basin/Lake | This study | UNEP (2013) | Awange et al. (2014) |
|---|---|---|---|
| Lake Victoria Basin | 256 100 | 184 000 | 258 000 |
| Lake Victoria | 67 220 | 68 800 | - |
| Lake Kyoga Basin | 79 270 | 75 000 | 75 000 |
| Lake Kyoga | 2 730 | 1 720 | - |




**Table 2.** Details of groundwater and lake level monitoring stations located in Lake Victoria
Basin and Lake Kyoga Basin.

| Monitoring Station | Basin | Parameter | Longitude | Latitude | Depth (m bgl) |
|---|---|---|---|---|---|
| Apac | LKB | Groundwater level | 32.50 | 1.99 | 15.0 |
| Pallisa | LKB | Groundwater level | 33.69 | 1.20 | 46.2 |
| Soroti | LKB | Groundwater level | 33.63 | 1.69 | 66.0 |
| Bugondo | LKB | Lake level | 33.20 | 0.45 | - |
| Entebbe | LVB | Groundwater level | 32.47 | 0.04 | 48.0 |
| Rakai | LVB | Groundwater level | 31.40 | −0.69 | 53.0 |
| Nkokonjeru | LVB | Groundwater level | 32.91 | 0.24 | 30.0 |
| Jinja | LVB | Lake level | 33.23 | 1.59 | - |




**Table 3.** Linear trends (cm yr$^{-1}$) in GRACE ΔTWS and in situ ΔTWS in Lake Victoria Basin and
Lake Kyoga Basin over various time periods (statistically significant trends, *p* values <0.05 are
marked by an asterisk).

| Period | GRACE Ensemble | GRGS | JPL-Mascons | In situ TWS |
|---|---|---|---|---|
| Lake Victoria Basin (LVB) | | | | |
| 2003−2006 | −4.10* | −9.00* | −7.70* | −11.00* |
| 2007−2012 | −0.31 | 1.50* | 1.90* | 1.10* |
| 2003−2012 | 0.04 | 0.58 | 0.79* | 0.54* |
| Lake Kyoga Basin (LKB) | | | | |
| 2003−2006 | −2.10* | −4.60* | −5.60* | −2.80* |
| 2007−2012 | 0.22 | 2.00* | 2.20* | 0.48 |
| 2003−2012 | −0.01 | 0.54* | 0.55* | −0.56* |

















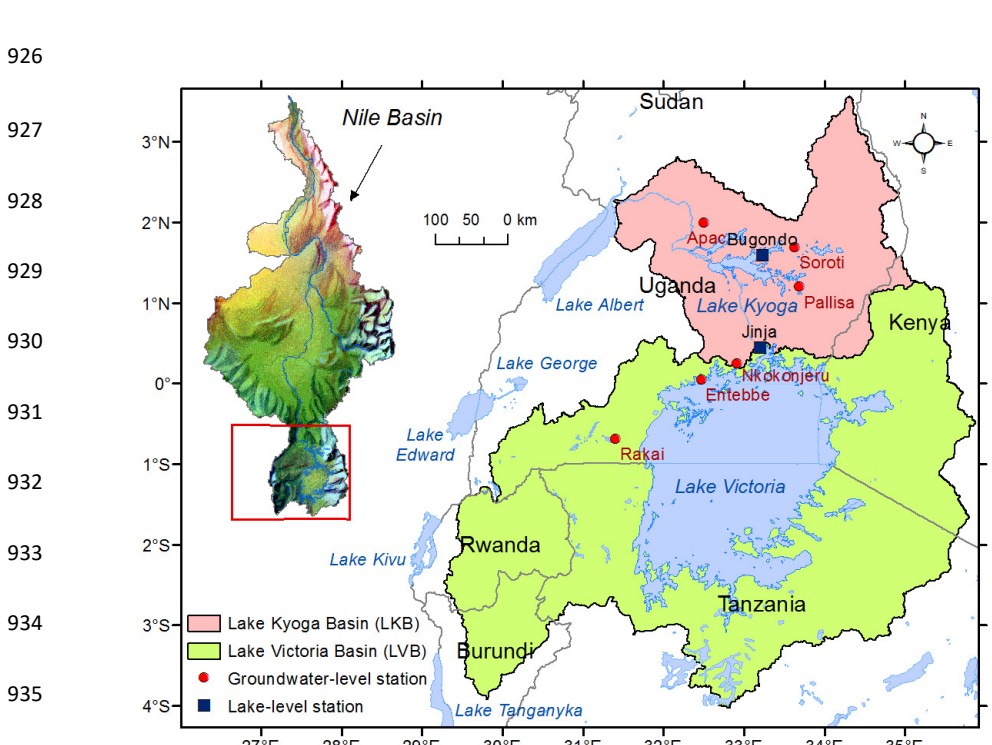

**Figure 1.** Map of the study area encompassing the Lake Victoria Basin (LVB) and Lake Kyoga
Basin (LKB), and location of the in situ monitoring stations. The Upper Nile Basin is marked by
a rectangle (red) within the entire Nile River Basin shown as a shaded relief index map.
Piezometric monitoring (red circles) and lake-level gauging (dark blue squares) stations are
shown on the map.
















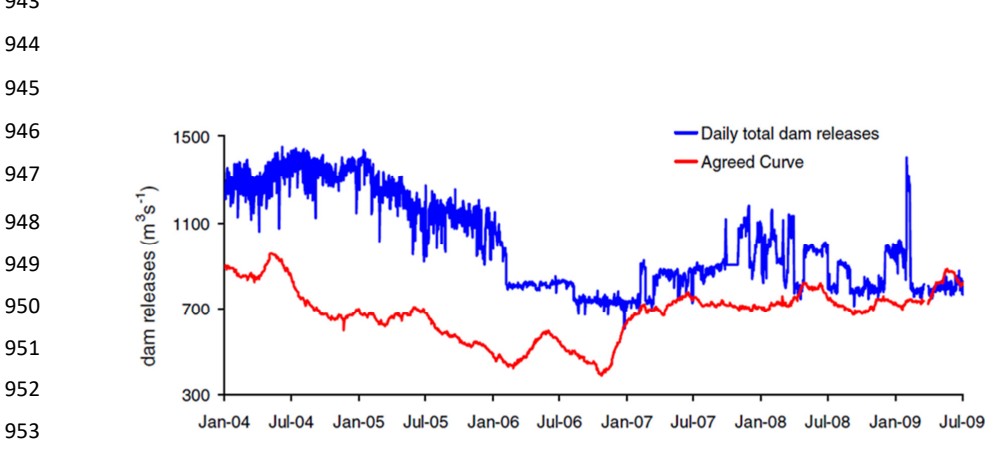

**Figure 2.** Observed daily total dam releases (blue line) and the agreed curve (red line) at the
outlet of Lake Victoria in Jinja from November 2007 to July 2009 (Owor et al., 2011).





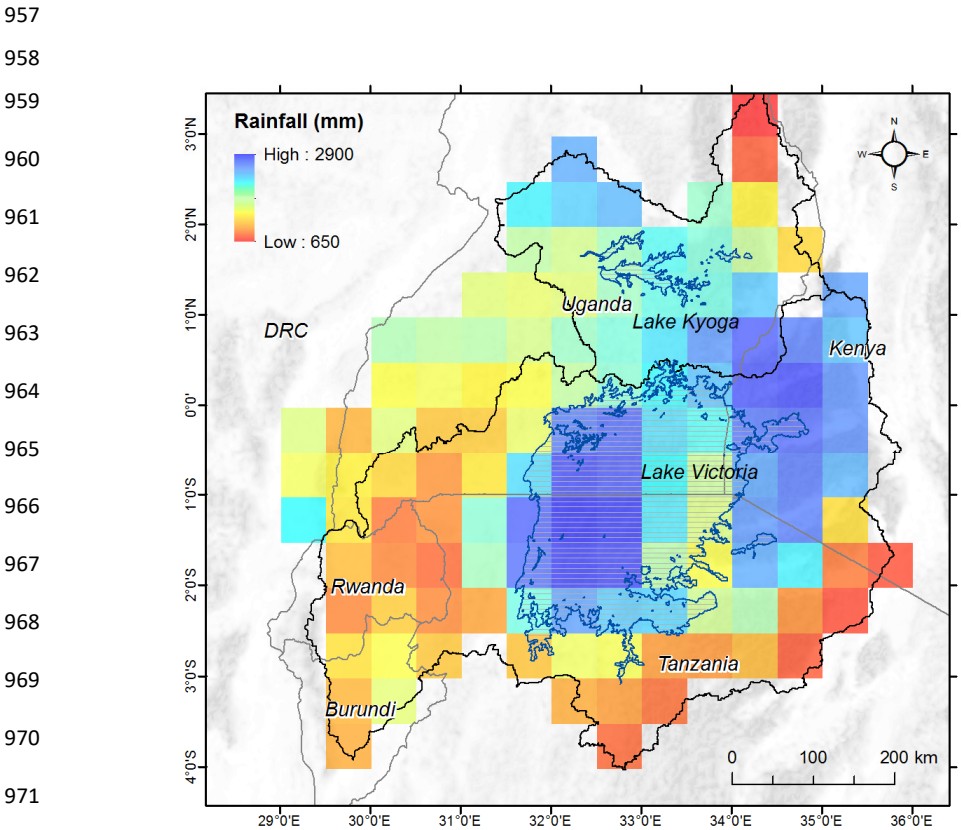

**Figure 3.** Mean annual rainfall for the period of 2003−2012 derived from TRMM satellite
observations. Greater annual rainfall is observed over much of the Lake Victoria and
northeastern corner of the Lake Victoria Basin.






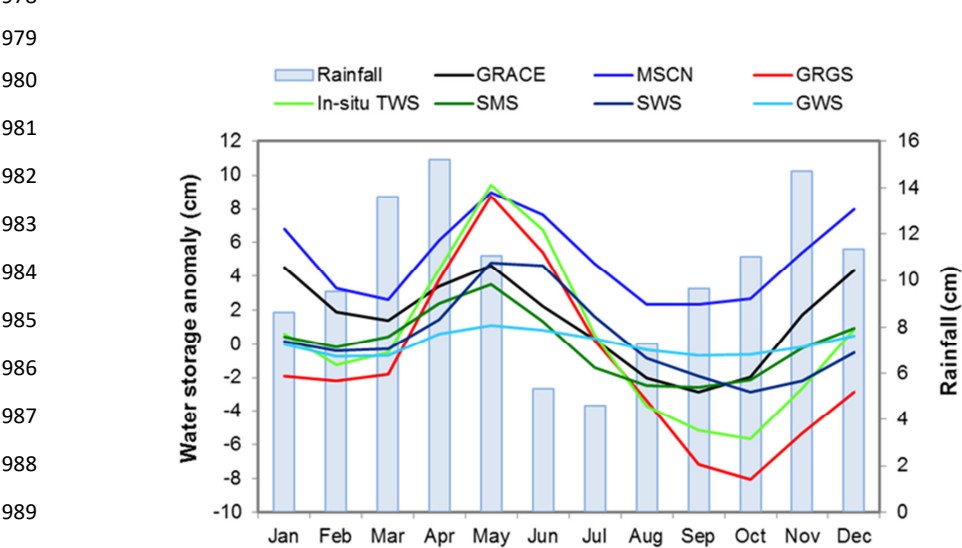

**Figure 4.** Seasonal pattern of TRMM-derived monthly rainfall, various GRACE-derived ΔTWS signals [GRCE=ensemble mean of CSR, GFZ, and JPL; GRGS and JPL-Mascons (MSCN) products], GLDAS LSMs ensemble ΔSMS, in situ ΔSWS and ΔGWS over the Lake Victoria Basin.

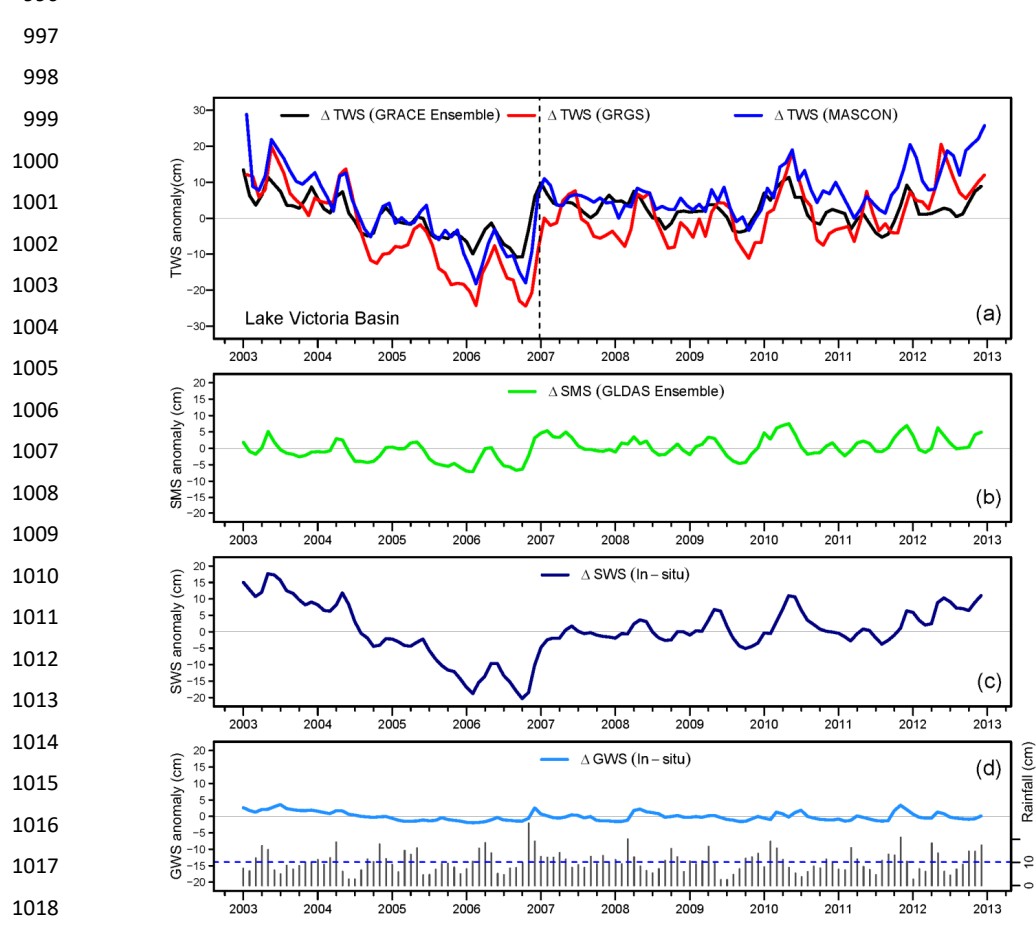

**Figure 5.** Monthly time-series datasets for the Lake Victoria Basin (LVB) from January 2003 to December 2012: (a) *GRCTellus* GRACE-derived ΔTWS (ensemble mean of CSR, GFZ, and JPL), GRGS and JPL-Mascons ΔTWS time-series data; (b) GLDAS-derived ΔSMS (ensemble mean of NOAH, CLM, and VIC); (c) lake-level-derived ΔSWS; and (d) borehole-derived ΔGWS time-series data.



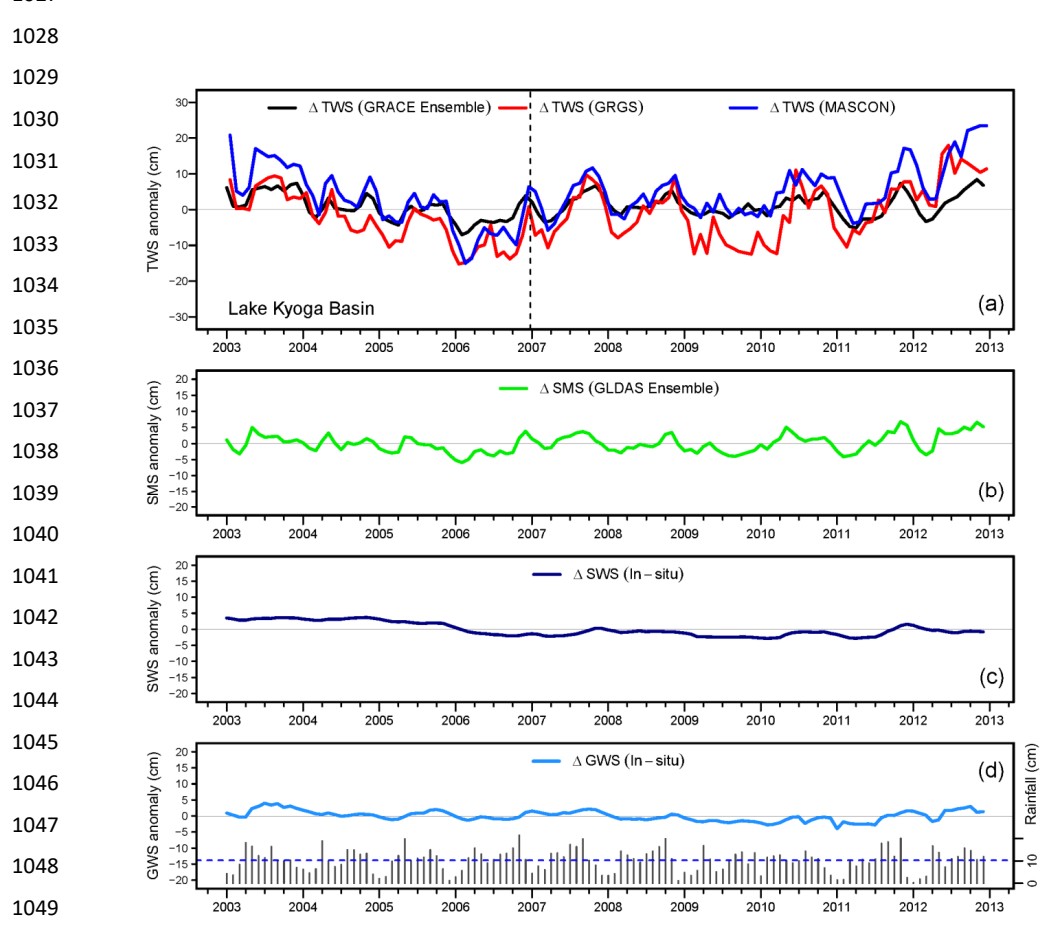

**Figure 6.** Monthly time-series datasets for the Lake Kyoga Basin (LKB) from January 2003 to December 2012: (a) *GRCTellus* GRACE-derived ΔTWS (ensemble mean of CSR, GFZ, and JPL), GRGS and JLP-Mascons ΔTWS time-series data; (b) GLDAS-derived ΔSMS (ensemble mean of NOAH, CLM, and VIC); (c) lake-level-derived ΔSWS; and (d) borehole-derived ΔGWS time-series data.


















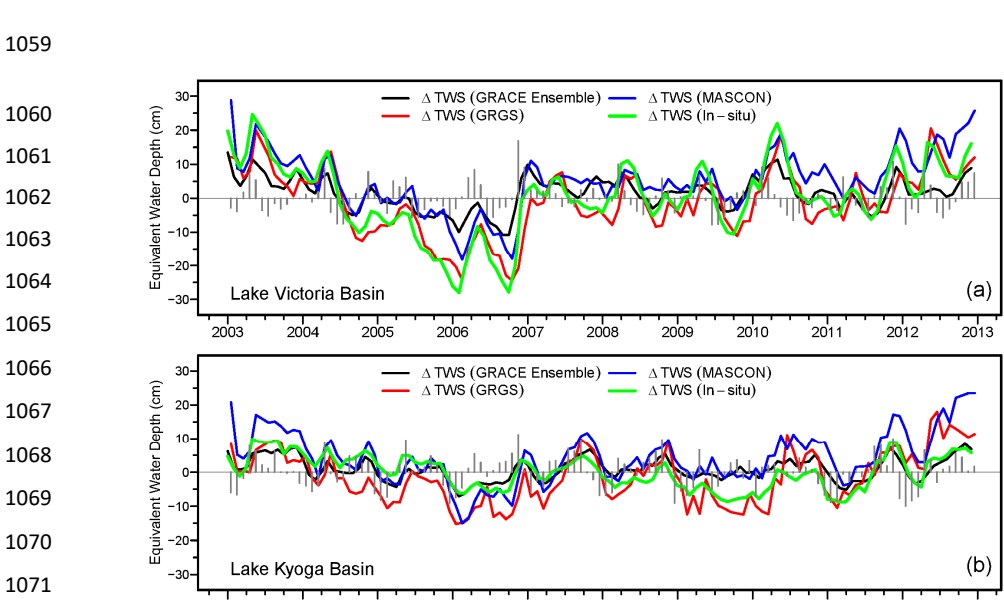

**Figure 7.** Comparison among time-series records of ΔTWS from *GRCTellus* (ensemble mean of
CSR, GFZ, and JPL), GRGS and JPL-MASCON GRACE products and in situ ΔTWS for the
Lake Victoria Basin (LVB) (a) and Lake Kyoga Basin (LKB), (b) for the period of 2003 to 2012.
The vertical grey lines represent monthly rainfall anomalies in LVB and LKB.







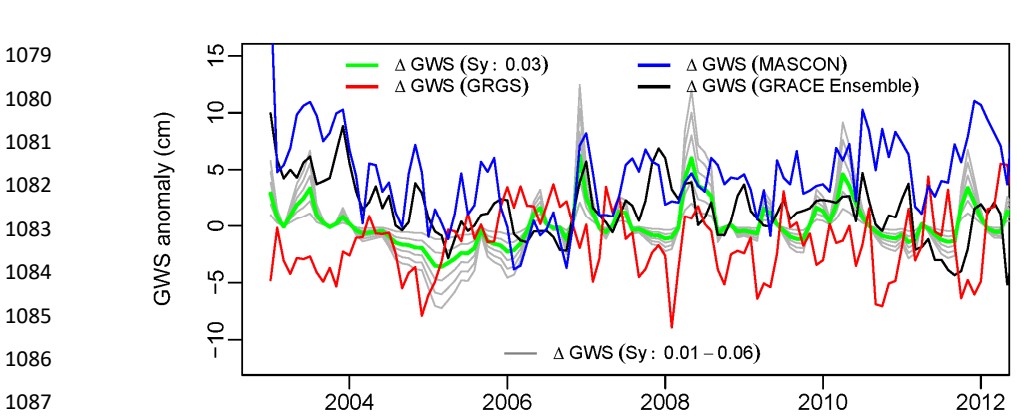


**Figure 8.** Estimates of in situ ΔGWS and GRACE-derived ΔGWS time-series records

(2003−2012) in LVB show a substantial variations among themselves. Note that an adjusted

ΔSWS (scaling factor of 0.11) is applied in the disaggregation of ΔGWS using *GRCTellus*

GRACE (ensemble mean of CSR, GFZ, and JPL) product; similarly, an adjusted ΔSWS (scaling

factor of 0.39) is applied for the JPL-Mascons product.

1094


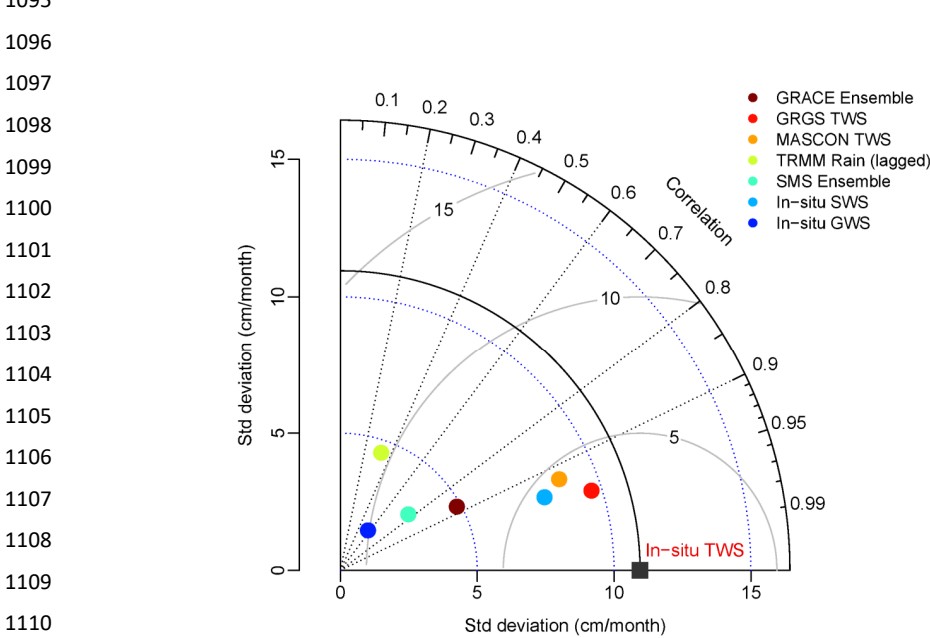

**Figure 9.** Taylor diagram shows strength of statistical association, variability in amplitudes of

time-series records and agreement among the reference data, in situ ΔTWS and *GRCTellus*

GRACE-derived ΔTWS (ensemble mean of CSR, GFZ, and JPL, GRGS and Mascons ΔTWS

time-series records), simulated ΔSMS (ensemble mean of NOAH, CLM, and VIC), in situ

ΔSWS, and in situ ΔGWS over the LVB. The solid arcs around the reference point (black

square) indicate cantered Root Mean Square (RMS) differences among in situ ΔTWS and other

variables, and the dashed arcs from the origin of the diagram indicate variability in time-series

records. Data for Lake Victoria Basin (LVB) are only shown in this diagram.