# Peer review of "Recent changes in terrestrial water storage in the Upper Nile Basin: an 1 evaluation of commonly used gridded GRACE products 2 3 Mohammad Shamsudduha1, 2, Richard G. Taylor2, Darren Jones3, Laurent 4 Longuevergne4, Michael Owor5 and Callist Tind"

_Hydrology and Earth System Sciences, 2017_

## Referee Comment (RC1) · Anonymous Referee #1 · 18 Apr 2017

This study aims to estimate the TWS change and its individual components in the Upper Nile Basin using GRACE, LSMs and in situ observations. Actually, similar studies have been done in this region by Awange et al. [2013], Awange et al. [2014], and Nanteza et al. [2016]. So, the main point is whether this manuscript can bring enough new knowledge based on new/updated data or methods. Different from previous studies, three different GRACE products (gridded level-3 GRCTellus, JPL mascon and constrained GRGS products) were compared and validated with in situ TWS observations in this study. However, the detailed scaling process used in this study is still unclear for me (see detailed comments below). I also suspect that whether limited 6 monitoring well observations can represent actual large-scale GWS variations in the study region.

Especially, all three well observations in the LVB are located near the Lake Victoria. The representativeness of these wells is questionable. In addition, there are some obvious typos in the manuscript.

GWS estimation from GRACE: Based on my understanding on the manuscript, $\Delta$GWS = the rescaled GRACE $\Delta$TWS (sf=1.7 for GRCTellus, sf=? for JPL mascon) minus scale-down $\Delta$SWS (sf=0.11 for GRCTellus and sf=0.39 for JPL mascon) minus simulated $\Delta$SMS. Why so-called a scale down of $\Delta$SWS was used rather than the original $\Delta$SWS (EWH, based on equation 2, Line 317)? In fact, the $\Delta$GWS estimation from GRACE (GRCTellus, JPL mascon and GRGS) was not given in detail. I would suggest the authors explain it in a paragraph in 3.2.2.

(1) Line 240, CRS should be CSR.

(2) Line248-249: GRCTellus datasets are provided as 1X1 grids, but $\sim$111 km is not the so-called spatial resolution of GRACE. At least, in some place of the manuscript, the authors should emphasize that the real resolution of GRACE is about 300 km, rather than that provided by these level-3 products.

(3) Line 254, the citation Geruo et al., 2013 should be A et al. 2013. This is also a mistake in some other papers. Actually, A is his family name and Geruo is his forename.

(4) Line 287, the citation (CSR, 2016) was not shown in the References. If there is no publication about it, maybe the authors can provide the website link where the information was available.

(5) Line 405, JLP should be JPL. For GRGS, whether scaling factor was applied?

(6) Line 310, Fig.s should be Figs.

(7) Line 394, if I understand it correctly, gridded scale factors from Landerer et al. were not used in this study finally. The authors applied a single scaling (1.7) actually. Based on Figure S1 and the authors' experiment (Fig. S10b), the factors are highly underestimated by Landerer et al. in the LVB.

(8) In 3.2 Methodologies, how to estimate GWS using GRACE in detail? I would suggest the authors explain it in a paragraph in 3.2.2.

(9) Line 434, "in both LVB and LKB (see supplementary Figs. S2–S7)." The captions of Figs. S5-S7 are "over the Victoria Nile Basin". Does the Victoria Nile Basin mean the LKB? The caption of Figure S9 also contains "in VNB".

(10) Line 436, simulated $\triangle$SWS should be simulated $\triangle$SMS?

(11) Line 446, "all 5 GRACE $\triangle$TWS and in situ $\triangle$TWS time-series records". There are only 4 curves in each panel of Figure 7.

(12) Line 449, "the period of 2004 to 2006", but in table 3, "2003-2006". This kind of inconsistency occurs several times in the manuscript.

(13) Line 464, "see supplementary Table S1". No correlation estimates in table S1 in fact.

(14) Line 465-466, "GRACE $\triangle$TWS is unable to explain natural variability in in situ $\triangle$TWS in LKB though this may be explained by the fact that SWS in Lake Kyoga is influenced by dam releases from LVB". GRACE can detect all mass changes including both natural and anthropogenic variability, but can not disaggregate individual components. If in situ $\triangle$TWS includes all mass change signals, it should be consistent with the GRACE estimate, no matter mass change is natural or anthropogenic. I suspect that the lower correlation in the LKB might be caused by the smaller area of LKB and larger leakage errors from the surrounding regions (including LVB).

(15) The caption of Table S1, no "variablility (i.e., variance, cmˆ2)" in the table. In the caption, what is the meaning of 120 cmˆ2 and 24 cmˆ2? The variances of in situ $\triangle$TWS?

(16) Line 473-477, GRACE-derived $\triangle$TWS was rescaled to recover the actual mass change. But, why the scaling down process was needed to remove $\triangle$SWS for estimating $\triangle$GWS? If rescaled $\triangle$TWS time series was used to estimate $\triangle$GWS, maybe the

authors should use in situ ΔSWS (equation 2) rather than scaling down ΔSWS. I also cannot understand the caption of Figure 8. Why a scaling down process of SWS is needed for disaggregating GWS from GRACE-derived rescaled TWS (Line 399-405)

(17) Line 399-405, were these factors calculated from the product of Landerer and Swenson 2012 (Figure S1)? Note that this product should be used for recovering TWS rather than only for SWS. In line 402, s=0.71 for experiment 1. But in caption of Figure S10, s=0.77 for experiment 1.

(18) Section 3.1.3, GLDAS does not assimilate surface water, which is an important TWS component in the study region. Whether the absence of surface water process will highly affect the accuracy of simulated soil moisture from GLDAS? Maybe the authors can try to use WGHM model which considers the surface water. In Figure S12, the authors compared many LSMs except WGHM, which simulates all TWS components. If the authors removed ΔSMS from WGHM, maybe there will be a better agreement between in situ well observations and GRACE-based ΔGWS, although the representativeness of these wells is also questionable.

(19) Line 1117, cantered should be centered.

(20) Figure 8, what is the criterion of selecting Sy?

---

## Referee Comment (RC2) · Anonymous Referee #2 · 18 Apr 2017

This study evaluates, for the Upper Nile Basin over the 2003-2012 period, several estimates of terrestrial water storage (TWS) as processed from the Gravity Recovery and Climate Experiment (GRACE) retrievals with in situ and model-derived estimates of its individual terms: surface water storage (SWS), soil moisture storage (SMS), and groundwater storage (GWS).

The authors reach interesting conclusions, namely 1) the pre-processing of GRACE greatly affects estimated annual TWS amplitude and, most notably, reconcilability with bottom-up approaches and 2) uncertainty in GRACE TWS and model-derived prevents a reasonable inference of GWS variation in these aquifers.

While I appreciate the scientific value of this work, I find this manuscript confusing at

times in its logic, and lacking rigor regarding how methods and some quantities are defined. Therefore, I recommend resubmission only after the authors have made a substantial rewriting effort to improve the clarity of the presented results.

**General comments**

- "In situ $\Delta$TWS" is used throughout the manuscript, but this term is quite misleading: as defined in Eq. (1) and then L379-381, this quantity is the sum of $\Delta$SWS, $\Delta$GWS, and $\Delta$SMS estimates. While the two former terms are indeed estimates based on situ measurements, $\Delta$SMS is averaged from simulations with three *gridded* hydrological models at 0.25° resolution (Sect 3.1.3 and L580-581). This is of particular importance since the whole study is about attempting to reconcile estimates of storage compartments across approaches and scales. I suggest using something like "bottom-up $\Delta$TWS" instead.

- The method section is rather long, in particular the description of GRACE datasets retrievals and the applied methodology in sections 3.1.2, 3.2.1 and 3.2.2. While I understand the authors want to present the remaining datasets ($\Delta$SWS, $\Delta$SMS, . . . ) before detailed how $\Delta$TWS is being processed, sect. 3.2.1 and sect 3.2.2, are even frankly confusing at times, e.g., when the $\Delta$TWS scaling methodology is explained (L357-363, see specific comments) and then discussed again (L387-397) so that in the end I am not sure what was used for the study.

- TWS sometimes appears instead of $\Delta$TWS (e.g. L79-86). While this be should a mere technical comment, in some cases TWS would actually be more accurate in the general sense (i.e. the concept of storage), e.g. when discussing reduction in volumetric storage in the whole basin (e.g., L537-539 where "$\Delta$TWS" is used).

**Specific comments**

**L21-22:** It would be more accurate to say that the authors *"test the phase and ampli-tude of three GRACE $\Delta$TWS estimates derived from 5 commonly-used gridded prod-ucts [...]"*.

**L123:** What is the actual time span of the "unintended experiment": 2004-2006 (like stated here)? 2005-2006 (e.g., L553)? 2003-2006 (most of the manuscript)? The authors should delimit this period consistently across the main text, the tables, the figures, and the supplementary materials.

**L169-173:** The authors should comment on the large discrepancy between these two lake area estimates. In addition, why do the authors report the *HydroSHEDS* area value as being from this study in Table 1?

**L357-363:** The authors first state that they spatially aggregate the unscaled $\Delta$TWS signal over the study region in order to have a time series, but then say that the scaling factors are applied to each grid of the GRACE mesh, therefore it is done before spatial aggregation? Please clarify.

**L395-397:** Along with the regionally-averaged gain factor, why did the authors not also test the third method described L392-394?

**L415-418:** A lag of 2-3 months between lowest rainfall and lowest $\Delta$TWS is also well noticeable, while $\Delta$SMS respond more quickly to rewetting after the driest month ($\approx$1 month) and $\Delta$SWS is slower ($\approx$4 months lag after minimum rainfall).

**L432-434:** Figs. S5 to S7 are relative to the entire Victoria Nile Basin and not Lake Kyoga Basin, I do not see how the authors can derive the observation that *"GRACE-*

*derived ∆TWS signals are strongly correlated in both LVB and LKB (see supplementary Figs. S2–S7)".* The same applies L441-444. Maybe the figures were unintentionally swapped with relative to LKB?

**L446-447:** This sentence is misleading since only 3 ∆TWS estimates are used shown, albeit derived from 5 different GRACE products.

**L449-456:** The authors might already mention that only ∆GWS shows an increase in 2005-2006, as later discussed in the Discussion section.

**L457-458:** A support supplementary figure with time series for LKB would help. Is it what Fig. S9 should have been (instead of describing the Victoria Nile Basin)? If so, the authors should add a reference to Fig. S9 here, and replace *"[. . . ] (see supplementary Figs. S8–S9)."* by *"[. . . ] (see supplementary Figs. S8–S9)."* in L456, and caption of Fig. S9 should read "LKB", instead of "VNB".

**L465-466:** I am not sure what the authors mean, how could the TWS signal miss one of its component, unless it refers to a water transfer within the system? All the more that even if mention of LVB-driven water balance of LKB is given on L175-177, this point is not picked up later in the Discussion section. Is it related to the substantial variability of ∆TWS deriving from ∆SMS in in LKB as compared to LVB? Could the authors expand their idea?

**L476-477:** Why scaling down ∆SWS rather than using the rescaled ∆TWS presented right above (L474-476) to disaggregate ∆GWS?

**L526-527:** This sentence essentially repeats L517-518, with typos (see *Technical comments*).

**L529:** The measurement error is not necessarily only a bias (systematic) is there are

random components; Swenson and Wahr (2006) seem to keep this broader definition.

**L541-548:** Would not it be more correct to say that the choice of $\Delta$SMS from LSMs contributes to uncertainty in estimating bottom-up $\Delta$TWS (termed in situ in the manuscript, see *General Comments*), and consequently comparing it to GRACE $\Delta$TWS, rather than uncertainty "GRACE analysis"? In addition, the order of sentences in this paragraph leaves me with the impression that this study did not bring any improvement to estimating bottom-up $\Delta$TWS, while most of the manuscript uses this estimate as a benchmark to test GRACE $\Delta$TWS products. In order to avoid finally leaving the reader with *"how reliable is this $\Delta$TWS reconciliation then?"*, the authors should maybe remind in the discussion that $\Delta$SWS is by far the largest contributor in LVB at least, somewhat limiting the propagation of $\Delta$SMS uncertainty.

**L616-617:** This should probably be stated already in the Discussion.

**Technical corrections**

**L101:** SSA is not used anywhere else in the manuscript of supplement.

**L527:** Likely typos, maybe *"[. . . ] priori information from LSMs contributes to adding uncertainty to $\Delta$TWS signals"*.

**Figs. 5 and 6:** What are the dashed vertical lines in the top panels and the horizontal dashed line in the bottom panels?

**References**

- Swenson, S., and Wahr, J.: Post-processing removal of correlated errors in GRACE data, Geophys. Res. Lett., 33, L08402, doi:10.1029/2005GL025285, 2006.

---

## Author Comment (AC1) · 3 May 2017

Response to Anonymous Referee 1 (AR1)

Numbered responses are given below each comment:

[AR1] This study aims to estimate the TWS change and its individual components in the Upper Nile Basin using GRACE, LSMs and in situ observations. Actually, similar studies have been done in this region by Awange et al. [2013], Awange et al. [2014], and Nanteza et al. [2016]. So, the main point is whether this manuscript can bring enough new knowledge based on new/updated data or methods. Different from previous studies, three different GRACE products (gridded level-3 GRCTellus, JPL mascon

and constrained GRGS products) were compared and validated with in situ TWS observations in this study. However, the detailed scaling process used in this study is still unclear for me (see detailed comments below). I also suspect that whether limited 6 monitoring well observations can represent actual large-scale GWS variations in the study region.

Responses to general comments [G1 to G3]:

[G1] We thank the Anonymous Referee #1 (AR1) for their comments on the manuscript. We are pleased that the reviewer has recognised the central difference between this study and previous studies in the region that include: (1) application of commonly used gridded GRACE products rather than a single GRACE product; and (2) an evaluation of these gridded products to represent the phase and amplitude of changes in terrestrial water storage in the Upper Nile Basin including a large and well-constrained change in surface water storage from 2003 to 2006.

[AR1] Especially, all three well observations in the LVB are located near the Lake Victoria. The representativeness of these wells is questionable. In addition, there are some obvious typos in the manuscript.

[G2] We agree with AR1 that the representivity of a limited number (6) of monitoring wells in the region is questionable. These daily monitoring records have been selected from a larger database of groundwater-level monitoring records in Uganda on the basis of the completeness and quality of their records from 2003 to 2012. Unfortunately, several time-series records from Uganda were excluded due to unexplained errors and substantial gaps; the location of the several monitoring wells also resided outside of the studied basins. Long time-series records of groundwater levels over the period from 2003 to 2012 from western Kenya, northern Tanzania, Rwanda and Burundi have not been identified despite intensive investigations carried out by The Chronicles Consortium, https://www.un-igrac.org/special-project/chronicles-consortium.

In the supplementary information of the revised manuscript, we will include plots of all

employed piezometric observations that inform in situ $\triangle$GWS (Fig. R1). In the Lake Kyoga Basin, piezometric records from 3 sites show consistency in the seasonality and amplitude of groundwater storage changes plotted as monthly groundwater-level anomalies relative to the mean for the period from 2003 to 2012; further details of these oscillations are described by Owor et al. (2009). In the Lake Victoria Basin, groundwater-level records from 2 sites (Entebbe, Nkokonjeru) are similar in their phase and amplitude, and are influenced by changes in the level of Lake Victoria as demonstrated by Owor et al. (2011). The groundwater-level record from Rakai represents local semi-arid conditions that exist within catchment areas (e.g. River Ruizi) draining to the western shore of Lake Victoria in Uganda. Although there are differences in the phase of groundwater-level fluctuations between the semi-arid site at Rakai and both Entebbe and Nkokonjeru (as well as the 3 sites in the Lake Kyoga Basin), amplitudes are similar.

[AR1] GWS estimation from GRACE: Based on my understanding on the manuscript, $\triangle$GWS= the rescaled GRACE $\triangle$TWS (sf=1.7 for GRCTellus, sf=? for JPL mascon) minus scale-down $\triangle$SWS (sf=0.11 for GRCTellus and sf=0.39 for JPL mascon) minus simulated $\triangle$SMS. Why so-called a scale down of $\triangle$SWS was used rather than the original $\triangle$SWS (EWH, based on equation 2, Line 317)? In fact, the $\triangle$GWS estimation from GRACE (GRCTellus, JPL mascon and GRGS) was not given in detail. I would suggest the authors explain it in a paragraph in 3.2.2.

[G3] We thank the AR1 for this critical comment on the estimation of $\triangle$GWS derived from GRACE datasets.

First, GRACE $\triangle$TWS time-series records were generated for LVB and LKB following a conventional approach by: (i) selecting $1° \times 1°$ grids within the basin boundary, (ii) applying gridded scaling factors to the corresponding $\triangle$TWS grids; and (iii) taking the average of time-series records of scaled $\triangle$TWS grids over the basin. For GRCTellus products (CSR, JPL, GFZ), we applied scaling coefficients derived from CLM4.0 land surface model provided by Landerer and Swenson (2012). Similarly, gridded scaling

factors were applied to JPL-Mascons product provided by Wiese et al. (2015). No scaling factors were applied to GRGS GRACE. On the specific question of 'rescaled GRACE ∆TWS', we did not apply a single multiplicative scaling factor of 1.7 to GRCTellus ∆TWS (CSR, JPL, GFZ products) to generate a basin-wide time-series data.

Two separate, unconventional scaling experiments were conducted in an attempt to reconcile GRCTellus TWS with in-situ (i.e. now 'bottom-up') TWS only for the Lake Victoria Basin (LVB). Under the first experiment, we applied a single multiplicative scaling factor of 1.7, informed by the lowest RMSE, in order to 'scale up" the GRCTellus ensemble mean of ∆TWS data. In the second experiment, we 'scaled down' SWS in the LVB, recognising that ∆SWS is the largest contributor to ∆TWS in the LVB. As stated on lines 399 to 404 in the current manuscript, "GRACE ∆TWS analyses commonly apply the same scaling factor as ∆TWS to all other individual components (Landerer and Swenson, 2012). We apply spatially-averaged scaling factors representative of (1) Lake Victoria and its surrounding grid cells (experiment 1: s=0.71; range 0.02−1.5), and (2) the open water surface of Lake Victoria without surrounding grid cells (experiment 2: s=0.11; range 0.02−0.30)."

To estimate ∆GWS from GRACE ∆TWS (following the conventional scaling approach outlined above), we applied the 'scaled down' SWS in the LVB because the amplitude of monthly anomalies of ∆SWS+∆SMS substantially exceed ∆TWS, particularly for the GRCTellus GRACE ∆TWS signal (Fig. R2 top). This discrepancy is pronounced over the period from 2003 to 2006, and produces steep, rising trends in the estimated GRACE-derived ∆GWS (i.e. GRACE ∆TWS - (∆SWS+∆SMS)) when borehole-derived (in situ) estimates of ∆GWS are declining and of much lower amplitude (Fig. R2 bottom).

We agree with AR1 that current description of application of scaling factors, both conventionally and unconventionally is insufficiently clear and will be substantially improved in the revised manuscript.

Responses to specific comments [S1 to S20]:

(1) Line 240, CRS should be CSR.

[S1] Agreed, to be corrected in revised manuscript.

(2) Line248-249: GRCTellus datasets are provided as 1X1 grids, but ∼111 km is not the so-called spatial resolution of GRACE. At least, in some place of the manuscript, the authors should emphasize that the real resolution of GRACE is about 300 km, rather than that provided by these level-3 products.

[S2] We thank AR1 for their point of clarity on the resolution of GRACE products. In the revised manuscript, we will state clearly the spatial resolution of GRACE footprint (∼300 km) in section 3.1.2.

(3) Line 254, the citation Geruo et al., 2013 should be A et al. 2013. This is also a mistake in some other papers. Actually, A is his family name and Geruo is his forename.

[S3] Agreed, to be corrected in revised manuscript.

(4) Line 287, the citation (CSR, 2016) was not shown in the References. If there is no publication about it, maybe the authors can provide the website link where the information was available.

[S4] Agreed, to be corrected in revised manuscript.

(5) Line 405, JLP should be JPL. For GRGS, whether scaling factor was applied?

[S5] Agreed, JPL will be corrected in revised manuscript; no scaling factors were applied for GRGS product.

(6) Line 310, Fig.s should be Figs. [S6] Agreed, to be corrected in revised manuscript.

(7) Line 394, if I understand it correctly, gridded scale factors from Landerer et al. were not used in this study finally. The authors applied a single scaling (1.7) actually. Based on Figure S1 and the authors' experiment (Fig. S10b), the factors are highly

underestimated by Landerer et al. in the LVB.

[S7] We applied the gridded scaling factors, provided by Landerer and Swenson (2012) to derive the GRACE $\Delta$TWS time-series data in the two basins. Under additional scaling experiments applied solely to GRCTellus, a single, multiplicative scaling factor of 1.7 was applied in an attempt to reconcile large differences between GRACE $\Delta$TWS and the in-situ (now 'bottom-up') $\Delta$TWS.

(8) In 3.2 Methodologies, how to estimate GWS using GRACE in detail? I would suggest the authors explain it in a paragraph in 3.2.2.

[S8] As per response G3, the estimation of $\Delta$GWS from GRACE will be stated explicitly in a revised section of 3.1.2.

(9) Line 434, "in both LVB and LKB (see supplementary Figs. S2–S7)." The captions of Figs. S5-S7 are "over the Victoria Nile Basin". Does the Victoria Nile Basin mean the LKB? The caption of Figure S9 also contains "in VNB".

[S9] Agreed, to be corrected in revised manuscript.

(10) Line 436, simulated $\Delta$SWS should be simulated $\Delta$SMS?

[S10] Agreed, to be corrected in revised manuscript.

(11) Line 446, "all 5 GRACE $\Delta$TWS and in situ $\Delta$TWS time-series records". There are only 4 curves in each panel of Figure 7.

[S11] Agreed, in the revised manuscript, the text will be revised to "all 3 $\Delta$TWS time-series records from 5 GRACE products".

(12) Line 449, "the period of 2004 to 2006", but in table 3, "2003-2006". This kind of inconsistency occurs several times in the manuscript.

[S12] Agreed, to be corrected as "2003 to 2006" throughout the revised manuscript.

(13) Line 464, "see supplementary Table S1". No correlation estimates in table S1 in

fact.

[S13] Agreed, we will correct the text in the revised manuscript. No correlation estimates are provided in the supplementary Table S1, which reports variances explained by linear regression.

(14) Line 465-466, "GRACE $\Delta$TWS is unable to explain natural variability in in situ $\Delta$TWS in LKB though this may be explained by the fact that SWS in Lake Kyoga is influenced by dam releases from LVB". GRACE can detect all mass changes including both natural and anthropogenic variability, but can not disaggregate individual components. If in situ $\Delta$TWS includes all mass change signals, it should be consistent with the GRACE estimate, no matter mass change is natural or anthropogenic. I suspect that the lower correlation in the LKB might be caused by the smaller area of LKB and larger leakage errors from the surrounding regions (including LVB).

[S14] We appreciate that GRACE detects all mass changes whether they are natural or anthropogenic. We also appreciate the explanation suggested by AR1 and provide calculations to support this assertion. In the current manuscript (lines 532 to 536), we briefly discuss the leakage from Lake Victoria into the adjacent basin but we will expand this discussion and report that our leakage analysis shows that GRACE signal leakage into LKB from LVB, which is 3 times larger, is 3.4 times bigger for both GRCTellus GRACE and GRGS products.

(15) The caption of Table S1, no "variablility (i.e., variance, cmËĘ2)" in the table. In the caption, what is the meaning of 120 cmËĘ2 and 24 cmËĘ2? The variances of in situ $\Delta$TWS?

[S15] Yes, the values of 120 and 24 cm^2 are variances in in-situ $\Delta$TWS for LVB and LKB respectively. The table legend will be correctly in the revised manuscript.

(16) Line 473-477, GRACE-derived $\Delta$TWS was rescaled to recover the actual mass change. But, why the scaling down process was needed to remove $\Delta$SWS for estimating ΔGWS? If rescaled ΔTWS time series was used to estimate ΔGWS, maybe the authors should use in situ ΔSWS (equation 2) rather than scaling down ΔSWS. I also cannot understand the caption of Figure 8. Why a scaling down process of SWS is needed for disaggregating GWS from GRACE-derived rescaled TWS (Line 399-405).

[S16] See above response to the general comment G3.

(17) Line 399-405, were these factors calculated from the product of Landerer and Swenson 2012 (Figure S1)? Note that this product should be used for recovering TWS rather than only for SWS. In line 402, s=0.71 for experiment 1. But in caption of Figure S10, s=0.77 for experiment 1.

[S17] Yes, as per response G3 and S7, gridded scaling factors provided by Landerer and Swenson (2012) were used to generate ΔTWS time-series records for LVB and LKB. However, under the scaling experiments undertaken on the ensemble mean of 3 GRCTellus GRACE products (CSR, JPL, GFZ), several multiplicative scaling factors were applied to observed ΔSWS time-series data (s=0.71 and s=0.11 in experiment 1) and ΔTWS (s=1.7 in experiment 2), guided by RMSE, in an attempt to reconcile substantial differences between GRCTellus GRACE ΔTWS and bottom-up ΔTWS.

(18) Section 3.1.3, GLDAS does not assimilate surface water, which is an important TWS component in the study region. Whether the absence of surface water process will highly affect the accuracy of simulated soil moisture from GLDAS? Maybe the authors can try to use WGHM model which considers the surface water. In Figure S12, the authors compared many LSMs except WGHM, which simulates all TWS components. If the authors removed ΔSMS from WGHM, maybe there will be a better agreement between in situ well observations and GRACE-based ΔGWS, although the representativeness of these wells is also questionable.

[S18] We thank AR1 for their suggestion of the use of WGHM to test GRACE-derived ΔGWS but this is beyond the scope of the current study and we will consider such experiments in future studies.

(19) Line 1117, cantered should be centered.

[S19] Agreed, to be corrected in revised manuscript.

(20) Figure 8, what is the criterion of selecting Sy?

[S20] The explanation for the applied range of Sy values is currently given in lines 334-338.

References:

Landerer, F. W., and Swenson, S. C.: Accuracy of scaled GRACE terrestrial water storage estimates, Water Resour. Res., 48, W04531, 2012.

Owor, M., Taylor, R. G., Tindimugaya, C., and Mwesigwa, D.: Rainfall intensity and groundwater recharge: empirical evidence from the Upper Nile Basin, Environmental Research Letters, 1-6, 2009.

Owor, M., Taylor, R. G., Mukwaya, C., and Tindimugaya, C.: Groundwater/surface-water interactions on deeply weathered surfaces of low relief: evidence from Lakes Victoria and Kyoga, Uganda, Hydrogeol. J., 19, 1403-1420, 2011.

Wiese, D. N., Yuan, D.-N., Boening, C., Landerer, F. W., and Watkins, M. M.: JPL GRACE Mascon Ocean, Ice, and Hydrology Equivalent Water Height JPL RL05M.1. Ver. 1, PO.DAAC, CA, USA, 2015.

[Figure]

[Figure]

[Figure]

**Fig. 1.** Figure R1. Time-series records of monthly anomaly of groundwater-level monitoring records at three stations in LVB (top), and records at three stations in LKB (bottom).

**Fig. 2.** Figure R2. Time-series records of GRACE $\Delta$TWS, sum of in-situ $\Delta$SWS and $\Delta$SMS, and in-situ $\Delta$GWS for LVB (top); and estimated $\Delta$GWS (bottom). Gridded scaling factors applied to GRCTellus and JPL-Mascons.

---

## Author Comment (AC2) · 3 May 2017

Response to Anonymous Referee 2 (AR2)

Numbered responses are given below each comment:

[AR2] This study evaluates, for the Upper Nile Basin over the 2003-2012 period, several estimates of terrestrial water storage (TWS) as processed from the Gravity Recovery and Climate Experiment (GRACE) retrievals with in situ and model-derived estimates of its individual terms: surface water storage (SWS), soil moisture storage (SMS), and groundwater storage (GWS). The authors reach interesting conclusions, namely 1) the pre-processing of GRACE greatly affects estimated annual TWS amplitude and, most

notably, reconcilability with bottom-up approaches and 2) uncertainty in GRACE TWS and model-derived prevents a reasonable inference of GWS variation in these aquifers. While I appreciate the scientific value of this work, I find this manuscript confusing at times in its logic, and lacking rigor regarding how methods and some quantities are defined. Therefore, I recommend resubmission only after the authors have made a substantial rewriting effort to improve the clarity of the presented results.

[G0] We greatly appreciate the critical comments of the Anonymous Referee #2 (AR2) and their recognition of the important conclusions of the manuscript.

Responses to general comments [G1-G3]

[AR2] "In situ $\Delta$TWS" is used throughout the manuscript, but this term is quite misleading: as defined in Eq. (1) and then L379-381, this quantity is the sum of $\Delta$SWS, $\Delta$GWS, and $\Delta$SMS estimates. While the two former terms are indeed estimates based on situ measurements, $\Delta$SMS is averaged from simulations with three gridded hydrological models at 0.25 resolution (Sect 3.1.3 and L580-581). This is of particular importance since the whole study is about attempting to reconcile estimates of storage compartments across approaches and scales. I suggest using something like "bottom-up $\Delta$TWS" instead.

[G1] We thank AR2 for their critical comment here. We agree and will adapt their proposed nomenclature, "bottom-up $\Delta$TWS", in the revised manuscript to make the distinction clearer.

[AR2] The method section is rather long, in particular the description of GRACE datasets retrievals and the applied methodology in sections 3.1.2, 3.2.1 and 3.2.2. While I understand the authors want to present the remaining datasets ($\Delta$SWS, $\Delta$SMS . . .) before detailed how $\Delta$TWS is being processed, sect. 3.2.1 and sect 3.2.2, are even frankly confusing at times, e.g., when the $\Delta$TWS scaling methodology is explained (L357-363, see specific comments) and then discussed again (L387-397) so that in the end I am not sure what was used for the study.

[G2] We appreciate that the description of various datasets and the method section are long and keep them separate under two sub-sections, Datasets (3.1) and Methodologies (3.2). The apparent confusion in the application of scaling factors may derive from the fact that we conducted additional scaling experiments only for the ensemble mean $\Delta$TWS of 3 GRCTellus GRACE products (CSR, JPL, GFZ). These additional scaling experiments were conducted in an attempt to reconcile GRCTellus GRACE $\Delta$TWS with 'bottom-up $\Delta$TWS'. As per responses S7 and S17 to AR1, we will clarify the selected methodologies for scaling factors in sections 3.2.1 and 3.2.2 in a revised manuscript.

[AR2] TWS sometimes appears instead of $\Delta$TWS (e.g. L79-86). While this be should a mere technical comment, in some cases TWS would actually be more accurate in the general sense (i.e. the concept of storage), e.g. when discussing reduction in volumetric storage in the whole basin (e.g., L537-539 where "$\Delta$TWS" is used).

[G3] We thank AR2 for their comment here and will revise the use of 'TWS' and '$\Delta$TWS' accordingly in a revised manuscript.

Responses to specific comments [S1 to S3]:

L21-22: It would be more accurate to say that the authors "test the phase and amplitude of three GRACE $\Delta$TWS estimates derived from 5 commonly-used gridded products [. . .]".

[S1] We thank AR2 for their critical comment and suggestion here. We agree with AR2 and will employ suggested edits in the revised manuscript.

L123: What is the actual time span of the "unintended experiment": 2004-2006 (like stated here)? 2005-2006 (e.g., L553)? 2003-2006 (most of the manuscript)? The authors should delimit this period consistently across the main text, the tables, the figures, and the supplementary materials.

[S2] Agreed, we will use the time span of 2003-2006 to indicate the "unintended experiment" throughout the revised manuscript.

L169-173: The authors should comment on the large discrepancy between these two lake area estimates. In addition, why do the authors report the HydroSHEDS area value as being from this study in Table 1?

[S3] We thank AR2 for their suggestion here and will include in a revised manuscript a statement highlighting the large discrepancy between the delineated area of LVB reported by UNEP (2013) and both Awange et al. (2014) and this study, which employs the HydroSHEDS boundary shapefiles for LVB and LKB.

L357-363: The authors first state that they spatially aggregate the unscaled $\Delta$TWS signal over the study region in order to have a time series, but then say that the scaling factors are applied to each grid of the GRACE mesh, therefore it is done before spatial aggregation? Please clarify.

[S4] Yes, gridded scaling factors were applied to corresponding grid cells for $\Delta$TWS before the spatial aggregation over LVB and LKB in order to generate time-series data. We will revise the texts in order to clarify this point.

L395-397: Along with the regionally-averaged gain factor, why did the authors not also test the third method described L392-394?

[S5] We do neither possess nor access monthly scaling factors to conduct the third scaling experiment and will clarify this point in the revised manuscript.

L415-418: A lag of 2-3 months between lowest rainfall and lowest $\Delta$TWS is also well noticeable, while $\Delta$SMS respond more quickly to rewetting after the driest month ($\sim$1 month) and $\Delta$SWS is slower ($\sim$4 months lag after minimum rainfall).

[S6] We appreciate this comment and will expand our discussion of seasonal hydrological responses to rainfall that include dam operations.

L432-434: Figs. S5 to S7 are relative to the entire Victoria Nile Basin and not Lake Kyoga Basin, I do not see how the authors can derive the observation that "GRACE-derived $\Delta$TWS signals are strongly correlated in both LVB and LKB (see supplemen-
tary Figs. S2–S7)". The same applies L441-444. Maybe the figures were unintention-ally swapped with relative to LKB?

[S7] Agreed, to be corrected in a revised manuscript.

L446-447: This sentence is misleading since only 3 $\Delta$TWS estimates are used shown, albeit derived from 5 different GRACE products.

[S8] Agreed, as per responses S11 (AR1) and S1 (AR2), we will revise the text.

L449-456: The authors might already mention that only $\Delta$GWS shows an increase in 2005-2006, as later discussed in the Discussion section.

[S9] We thank AR2 for this comment. We provide an explanation of the apparent rise in $\Delta$GWS in lines 554-557.

L457-458: A support supplementary figure with time series for LKB would help. Is it what Fig. S9 should have been (instead of describing the Victoria Nile Basin)? If so, the authors should add a reference to Fig. S9 here, and replace "[. . . ] (see supplementary Figs. S8–S9)." by "[. . . ] (see supplementary Figs. S8–S9)." in L456, and caption of Fig. S9 should read "LKB", instead of "VNB".

[S10] Agreed, LKB is mistakenly labelled as Victoria Nile Basin. We will correct this in a revised supplementary document.

L465-466: I am not sure what the authors mean, how could the TWS signal miss one of its component, unless it refers to a water transfer within the system? All the more that even if mention of LVB-driven water balance of LKB is given on L175-177, this point is not picked up later in the Discussion section. Is it related to the substantial variability of $\Delta$TWS deriving from $\Delta$SMS in in LKB as compared to LVB? Could the authors expand their idea?

[S11] We appreciate that GRACE detects all mass changes, whether they are natural or anthropogenic, and regret the confusion caused by our statement, "GRACE $\Delta$TWS

is unable to explain natural variability in in situ ΔTWS in LKB though this may be explained by the fact that SWS in Lake Kyoga is influenced by dam releases from LVB". As per response S14 (AR1), further discussion of signal leakage from Lake Victoria into Lake Kyoga will be made in the revised manuscript in which we will report on our leakage analysis showing that GRACE signal leakage into LKB from LVB, which is 3 times larger, is 3.4 times bigger for both GRCTellus GRACE and GRGS products.

L476-477: Why scaling down ΔSWS rather than using the rescaled ΔTWS presented right above (L474-476) to disaggregate ΔGWS?

[S12] As per response G3 to AR1, to estimate ΔGWS from GRACE ΔTWS, we applied a 'scaled down' SWS in the LVB because the amplitude of monthly anomalies of ΔSWS+ΔSMS substantially exceed ΔTWS, particularly for the GRCTellus GRACE ΔTWS signal (Fig. R1 top). This discrepancy is pronounced over the period from 2003 to 2006, and produces steep, rising trends in the estimated GRACE-derived ΔGWS (i.e. GRACE ΔTWS - (ΔSWS+ΔSMS)) when borehole-derived (in situ) estimates of ΔGWS are declining and of much lower amplitude (Fig. R1 bottom). We agree with AR2 that current description of application of scaling factors, both conventionally and unconventionally is insufficiently clear and will be substantially improved in the revised manuscript.

L526-527: This sentence essentially repeats L517-518, with typos (see Technical comments).

[S13] Agreed, lines 526-527 will be deleted in a revised manuscript.

L529: The measurement error is not necessarily only a bias (systematic) is there are random components; Swenson and Wahr (2006) seem to keep this broader definition.

[S14] We applied measurement and leakage errors from Landerer and Swenson (2012); reference to Swenson and Wahr (2006) is incorrect and will be corrected in the revised manuscript.

L541-548: Would not it be more correct to say that the choice of $\Delta$SMS from LSMs contributes to uncertainty in estimating bottom-up $\Delta$TWS (termed in situ in the manuscript, see General Comments), and consequently comparing it to GRACE $\Delta$TWS, rather than uncertainty "GRACE analysis"? In addition, the order of sentences in this paragraph leaves me with the impression that this study did not bring any improvement to estimating bottom-up $\Delta$TWS, while most of the manuscript uses this estimate as a benchmark to test GRACE $\Delta$TWS products. In order to avoid finally leaving the reader with "how reliable is this $\Delta$TWS reconciliation then?", the authors should maybe remind in the discussion that $\Delta$SWS is by far the largest contributor in LVB at least, somewhat limiting the propagation of $\Delta$SMS uncertainty.

[S15] We agree with this argument of AR2 that $\Delta$SWS is by far the largest contributor to $\Delta$TWS in the LVB and is dominated by an accurately observed $\Delta$SWS signal of 81 km3, limiting the propagation of $\Delta$SMS uncertainty. We will consequently revise the discussion to reflect this important argument as it relates to statements about uncertainty in GRACE products relative to a 'bottom-up' $\Delta$TWS.

L616-617: This should probably be stated already in the Discussion.

[S16] We thank AR2 for this suggestion.

Technical corrections:

L101: SSA is not used anywhere else in the manuscript of supplement.

[T1] Agreed, "(SSA)" will be deleted in a revised manuscript.

L527: Likely typos, maybe "[. . . ] priori information from LSMs contributes to adding uncertainty to $\Delta$TWS signals".

[T2] Agreed, this statement will be deleted in a revised manuscript.

Figs. 5 and 6: What are the dashed vertical lines in the top panels and the horizontal dashed line in the bottom panels?

[T3] Agreed, we will delete the vertical line which separates the two periods (2003-2006, 2007-2012). The dashed horizontal line indicates the mean rainfall for the period of 2003-2012; this detail will be made clear in the figure captions (Figures 5 and 6) in the revised manuscript.

―――――――――――――――――

[Figure]

Fig. 1. Figure R1. Time-series records of GRACE ΔTWS, sum of in-situ ΔSWS and ΔSMS, and in-situ ΔGWS for LVB (top); and estimated ΔGWS (bottom). Gridded scaling factors applied to GRCTellus and JPL-Mascons.

---

## Author Response (AR1)

**Anonymous Referee #1 [italics]**

**Numbered responses are given below each comment (see revised manuscript with track-changes accepted):**

This study aims to estimate the TWS change and its individual components in the Upper Nile Basin using GRACE, LSMs and in situ observations. Actually, similar studies have been done in this region by Awange et al. [2013], Awange et al. [2014], and Nanteza et al. [2016]. So, the main point is whether this manuscript can bring enough new knowledge based on new/updated data or methods. Different from previous studies, three different GRACE products (gridded level-3 GRCTellus, JPL mascon and constrained GRGS products) were compared and validated with in situ TWS observations in this study. However, the detailed scaling process used in this study is still unclear for me (see detailed comments below). I also suspect that whether limited 6 monitoring well observations can represent actual largescale GWS variations in the study region.

**Responses to general comments [G1 to G3]:**

**[G1]** We thank the Anonymous Referee #1 (AR1) for their comments on the manuscript. We are pleased that the reviewer has recognised the central difference between this study and previous studies in the region that include: (1) application of commonly used gridded GRACE products rather than a single GRACE product; and (2) an evaluation of these gridded products to represent the phase and amplitude of changes in terrestrial water storage in the Upper Nile Basin including a large and well-constrained change in surface water storage from 2003 to 2006.

Especially, all three well observations in the LVB are located near the Lake Victoria. The representativeness of these wells is questionable. In addition, there are some obvious typos in the manuscript.

**[G2]** We agree with AR1 that the representivity of a limited number (6) of monitoring wells in the region is questionable. These daily monitoring records have been selected from a larger database of groundwater-level monitoring records in Uganda on the basis of the completeness and quality of their records from 2003 to 2012. Unfortunately, several timeseries records from Uganda were excluded due to unexplained errors and substantial gaps; locations of the several monitoring wells also reside outside of the studied basins. Long timeseries records of groundwater levels over the period from 2003 to 2012 from western Kenya, northern Tanzania, Rwanda and Burundi have not been identified despite intensive investigations carried out by *The Chronicles Consortium*, https://www.un-igrac.org/special-project/chronicles-consortium.

In the supplementary information of the revised manuscript, we now include a new figure (see supplementary Fig. 3) showing the 6 employed piezometric records that inform in situ  $\Delta$ GWS (also see Fig. AR1.1). In the Lake Kyoga Basin, piezometric records from 3 sites show consistency in the seasonality and amplitude of groundwater storage changes plotted as monthly groundwater-level anomalies relative to the mean for the period from 2003 to 2012; further details of these oscillations are described by Owor et al. (2009). In the Lake Victoria Basin, groundwater-level records from 2 sites (Entebbe, Nkokonjeru) are similar in their phase and amplitude, and are influenced by changes in the level of Lake Victoria as demonstrated by Owor et al. (2011). The groundwater-level record from Rakai represents local semi-arid conditions that exist within catchment areas (e.g. River Ruizi) draining to the

western shore of Lake Victoria in Uganda. Although there are differences in the phase of groundwater-level fluctuations between the semi-arid site at Rakai and both Entebbe and Nkokonjeru (as well as the 3 sites in the Lake Kyoga Basin), amplitudes are similar.

Figure AR1.1. Time-series records of monthly anomaly of groundwater-level monitoring records at three stations in LVB (top), and records at three stations in LKB (bottom).

GWS estimation from GRACE: Based on my understanding on the manuscript,  $\Delta$ GWS= the rescaled GRACE  $\Delta$ TWS (sf=1.7 for GRCTellus, sf=? for JPL mascon) minus scale-down  $\Delta$ SWS (sf=0.11 for GRCTellus and sf=0.39 for JPL mascon) minus simulated  $\Delta$ SMS. Why so-called a scale down of  $\Delta$ SWS was used rather than the original  $\Delta$ SWS (EWH, based on equation 2, Line 317)? In fact, the  $\Delta$ GWS estimation from GRACE (GRCTellus, JPL mascon and GRGS) was not given in detail. I would suggest the authors explain it in a paragraph in 3.2.2.

**[G3]** We thank the AR1 for this critical comment on the estimation of  $\triangle$ GWS derived from GRACE datasets.

First, GRACE  $\Delta$ TWS time-series records were generated for LVB and LKB following a conventional approach by: (i) selecting 1° × 1° grids within the basin boundary, (ii) applying gridded scaling factors to the corresponding  $\Delta$ TWS grids; and (iii) taking the average of time-series records of scaled  $\Delta$ TWS grids over the basin. For *GRCTellus* products (CSR, JPL, GFZ), we applied scaling coefficients derived from CLM4.0 land surface model provided by Landerer and Swenson (2012). Similarly, gridded scaling factors were applied to JPL-Mascons product provided by Wiese et al. (2015). No scaling factors were applied to GRGS GRACE. On the specific question of 'rescaled GRACE  $\Delta$ TWS', we did not apply a single multiplicative scaling factor of 1.7 to *GRCTellus*  $\Delta$ TWS (CSR, JPL, GFZ products) to generate a basin-wide time-series data.

Below we explain why we conducted the scaling experiments. Please note, however, that in the revised manuscript, we now only apply an unscaled (original)  $\Delta$ SWS signal to estimate  $\Delta$ GWS from GRACE data. To respond to the original guery of the reviewer, we now provide a clearer explanation as to why unconventional scaling experiments were conducted (revised section 3.2.3). Two separate, unconventional scaling experiments were conducted only for the Lake Victoria Basin (LVB) in order to highlight the discrepancy between GRCTellus TWS and in-situ (i.e. now 'bottom-up') TWS and to reconcile GRCTellus with insitu observations since we observe that the amplitude of monthly anomalies of combined  $\Delta$ SWS+ $\Delta$ SMS signals substantially exceeds GRACE  $\Delta$ TWS, particularly for the *GRCTellus* GRACE  $\Delta$ TWS signal (see Fig. AR1.2 and supplementary Fig. S4). Under the first experiment, we apply a single multiplicative scaling factor of 1.7, informed by the lowest RMSE, in order to 'scale up' the *GRCTellus* ensemble mean of  $\Delta$ TWS data. In the second experiment, we apply a 'scaled down'  $\Delta$ SWS in the LVB, recognising that  $\Delta$ SWS is the largest contributor to  $\Delta TWS$  in the LVB. We apply spatially-averaged scaling factors representative of (1) Lake Victoria and its surrounding grid cells (experiment 1: s=0.71; range 0.02–1.5), and (2) the open water surface of Lake Victoria without surrounding grid cells (experiment 2: s=0.11; range 0.02-0.30)." These experiments suggest that the 'true' *GRCTellus* GRACE  $\Delta$ TWS signal is weakened during processing and that the combined  $\Delta$ SWS+ $\Delta$ SMS signal is greater than  $\Delta$ TWS, mathematically resulting to a positive estimate of  $\Delta GWS$ .

---

## Author Response (AR2)

10th August 2017

Chief Executive Editor
*Hydrology and Earth System Sciences*
Copernicus Publications

Manuscript Editor: Professor Ying Fan

Subject: Implementation of minor technical edits to a revised manuscript [hess-2017-146].

Dear Editor,

My co-authors and I are pleased to hear the decision of accepting the manuscript **"Recent changes in terrestrial water storage in the Upper Nile Basin: an evaluation of commonly used gridded GRACE products"** for publication in the *Hydrology and Earth System Sciences* journal with technical revisions.

We appreciate that one referee has raised concerns on the scientific originality and potential impact of the key findings of the paper and, therefore, you have asked to edit the manuscript in order to bring out the uniqueness of the work which will increase the long-term impact of the paper. We thank you for the positive decision and suggestions for technical edits to the current version of the manuscript. We have now slightly revised the manuscript by (1) referring to the recent global-scale studies (Scanlon et al., 2016 and Long et al., 2017), and (2) highlighting the key differences between our analyses of 5 GRACE products in the Upper Nile Basin and the two regional studies (Awange et al., 2014 and Nanteza et al., 2016) that applied a single GRACE product. New edits to the manuscript sections: Abstract, Introduction, Discussion and Conclusions can be found in the track-change (red texts) version of the revised manuscript.

We sincerely hope that you are satisfied with the technical revision of the manuscript and that the manuscript will be published in HESS soon.

Many thanks for your kind consideration.

Sincerely,

Dr. Mohammad Shamsudduha

UCL Institute for Risk & Disaster Reduction
University College London, Gower Street, London  WC1E 6BT
Tel:  +44 (0)20 3108 1103, E-mail: m.shamsudduha@ucl.ac.uk

[revised manuscript text omitted]

Long, D., Pan, Y., Zhou, J., Chen, Y., Hou, X., Hong, Y., Scanlon, B. R., and Longuevergne,

L.: Global analysis of spatiotemporal variability in merged total water storage changes using multiple GRACE products and global hydrological models, Remote Sensing of

Environment, 192, 198-216, 2017.

Longuevergne, L., Scanlon, B. R., and Wilson, C. R.: GRACE hydrological estimates for small basins: evaluating processing approaches on the High Plains Aquifer, USA,

Water Resour. Res., 46, W11517, 2010.

Longuevergne, L., Wilson, C. R., Scanlon, B. R., and Crétaux, J. F.: GRACE water storage
 estimates for the Middle East and other regions with significant reservoir and lake
 storage, Hydrol. Earth Syst. Sci., 17, 4817-4830, doi:10.5194/hess-17-4817-2013,
 2013.

MacDonald, A. M., Bonsor, H. C., Dochartaigh, B. E. O., and Taylor, R. G.: Quantitative
 maps of groundwater resources in Africa, Environ. Res. Lett., 7, doi:10.1088/1748-
 9326/1087/1082/024009, 2012.

Nanteza, J., de Linage, C. R., Thomas, B. F., and Famiglietti, J. S.: Monitoring groundwater
 storage changes in complex basement aquifers: An evaluation of the GRACE satellites
 over East Africa, Water Resour. Res., 52, doi:10.1002/2016WR018846, 2016.

Nicholson, S. E., Yin, X., and Ba, M. B.: On the feasibility of using a lake water balance
 model to infer rainfall: an example from Lake Victoria, Hydrological Science Journal,
 45, 75-95, 2000.

Niu, G.-Y., Yang, Z.-L., Dickinson, R. E., Gulden, L. E., and Su, H.: Development of a
 simple groundwater model for use in climate models and evaluation with Gravity
 Recovery and Climate Experiment data, J. Geophys. Res., 112, D07103,
 doi:10.1029/2006JD007522, 2007.

Oleson, K. W., Niu, G.-Y., Yang, Z.-L., Lawrence, D. M., Thornton, P. E., Lawrence, P. J.,
 Stockli, R., Dickinson, R. E., Bonan, G. B., Levis, S., Dai, A., and Qian, T.:
 Improvements to the Community Land Model and their impact on the hydrological
 cycle, J. Geophys. Res., 113, G01021, doi:10.1029/2007JG000563, 2008.

Owor, M., Taylor, R. G., Tindimugaya, C., and Mwesigwa, D.: Rainfall intensity and
 groundwater recharge: empirical evidence from the Upper Nile Basin, Environmental
 Research Letters, 1-6, 2009.

Owor, M.: Groundwater - surface water interactions on deeply weathered surfaces of low
 relief in the Upper Nile Basin of Uganda, Ph.D., Geography, University College
 London, London, 271 pp., 2010.

Owor, M., Taylor, R. G., Mukwaya, C., and Tindimugaya, C.: Groundwater/surface-water
 interactions on deeply weathered surfaces of low relief: evidence from Lakes Victoria
 and Kyoga, Uganda, Hydrogeol. J., 19, 1403-1420, 2011.

Ramillien, G., Famiglietti, J. S., and Wahr, J.: Detection of Continental Hydrology and
Glaciology Signals from GRACE: A Review, Surv. Geophys., 29, 361-374, 2008.

Rodell, M., and Famiglietti, J. S.: Terrestrial Water Storage Variations over Illinois: Analysis
of Observations and Implications for GRACE, Wat. Resour. Res., 37, 1327-1340, 2001.

Rodell, M., Houser, P. R., Jambor, U., Gottschalck, J., Mitchell, K., Meng, C.-J., Arsenault,
K., Cosgrove, B., Radakovich, J., Bosilovich, M., Entin, J. K., Walker, J. P., Lohmann,
D., and Toll, D.: The Global Land Data Assimilation System, Bull. Am. Meteorol. Soc.,
85, 381-394, 2004.

Rodell, M., Velicogna, I., and Famiglietti, J. S.: Satellite-based estimates of groundwater
depletion in India, Nature, 460, 999-1003, doi:10.1038/nature08238, 2009.

Rowlands, D. D., Luthcke, S. B., McCarthy, J. J., Klosko, S. M., Chinn, D. S., Lemoine, F.
G., Boy, J.-P., and Sabaka, T. J.: Global mass fluxsolutions from GRACE: A
comparison of parameter estimation strategies-Mass concentrations versus stokes
coefficients, J. Geophys. Res., 115, B01403, doi:10.1029/2009JB006546, 2010.

Scanlon, B. R., Longuevergne, L., and Long, D.: Ground referencing GRACE satellite
estimates of groundwater storage changes in the California Central Valley, USA, Water
Resour. Res., 48, W04520, 2012.

Scanlon, B. R., Zhang, Z., Reedy, R. C., Pool, D. R., Save, H., Long, D., Chen, J., Wolock,
D. M., Conway, B. D., and Winester, D.: Hydrologic implications of GRACE satellite
data in the Colorado River Basin, Water Resour. Res., 51, 9891-9903,
doi:10.1002/2015WR018090, 2015.

Scanlon, B. R., Zhang, Z., Save, H., Wiese, D. N., Landerer, F. W., Long, D., Longuevergne,
L., and Chen, J.: Global evaluation of new GRACE mascon products for hydrologic
applications, Water Resour. Res., 52, 9412-9429, doi:10.1002/2016WR019494., 2016.

[revised manuscript text omitted]